# AdaThink-Med: Medical Adaptive Thinking with Uncertainty-Guided Length Calibration

## Abstract

Recent advances in inference-time scaling with extended long chain-of thought have significantly improved the reasoning capabilities of both general and medical large language models (LLMs). However, these models tend to engage in lengthy reasoning processes regardless of the difficulty of the input question, leading to increased inference costs in real-world applications. Therefore, enabling adaptive thinking where models think less for simpler questions and think more for complex ones is critical for the effective use of medical LLMs in practice. Despite its importance, there is a lack of end-to-end approaches designed to enhance the adaptive thinking capabilities of medical LLMs while providing a comprehensive examination of the trade-off between performance and computational cost. To bridge this gap, we propose **AdaThink-Med**, the first end-to-end framework designed to enhance adaptive thinking ability in medical reasoning models with uncertainty-guided length calibration. Specifically, we propose a specific entropy-based difficulty estimator and plug it into a difficulty-aware length reward inside reinforcement fine-tuning, targeting medical reasoning models, and show emergent non-thinking versus thinking modes plus dataset-selection benefits. AdaThink-Med first generates multiple candidate outputs for each question, evaluates the correctness and uncertainty of each candidate, and then estimates problem difficulty via an uncertainty-guided length calibration module. For outputs with low difficulty and correct answers, the framework penalizes longer reasoning paths; whereas for those with high difficulty and incorrect answers, it encourages extending the chain of thought to explore alternative solutions. On six public medical QA benchmarks, AdaThink-Med achieves up to $6.4\times$ length reduction on average while retaining performance with only minimal degradation. Intriguingly, we observe that AdaThink-Med spontaneously develops two distinct reasoning modes, which we characterize as "non-thinking" and "thinking", demonstrating the model's ability to suppress redundant reasoning processes dynamically.[1]

## 1 Introduction

Recent advances in Chain-of-Thought (CoT) reasoning (Wei et al., 2022) and reinforcement learning (Suzgun et al., 2022; Zhou et al., 2022; Chen et al., 2024a; Guo et al., 2025) have significantly improved the reasoning abilities of LLMs with leveraging extended trial-and-error processes and reflective thinking—also known as inference time scaling. Such progress has inspired new paradigms for building general-purpose medical AI systems. Through high-quality long CoT data for supervised fine-tuning or rule-based reinforcement learning, substantial gains have been made in complex clinical tasks including differential diagnosis, surgical planning, and prognosis prediction (Chen et al., 2024b; Huang et al., 2025; Liu et al., 2025a; Rui et al., 2025b;a).

However, such improvements often come at the cost of significantly increased lengthy outputs, thereby raising inference time and costs. For example, when answering a simple question like "1 + 1 = ?", models such as DeepSeek-R1 (Guo et al., 2025) have been observed to produce outputs as long as 170 tokens. In parallel, another line of research has identified the *overthinking* phenomenon (Chen et al., 2024c; Luo et al., 2025), where reasoning models generate unnecessarily long chains of thought, which can even degrade performance on simple questions (Su et al., 2025).

---

[1]Code and model weights are available at: https://anonymous.4open.science/r/5829/.

Moreover, (Zhang et al., 2025) reports that up to 58% of questions in existing medical benchmarks can be accurately answered with short, concise outputs—without resorting to expensive extended reasoning. These findings collectively underscore the urgent need to improve the efficiency of reasoning models, especially if they are to be deployed effectively in real-world applications.

Most existing work aims to improve the efficiency of general reasoning models by applying length-based reward, which penalizes outputs with long chains of thought (CoTs). Specifically, (Team et al., 2025; Yu et al., 2025) encourages shorter outputs across the board, while another line of research (Shen et al., 2025; Yi et al., 2025; Yeo et al., 2025) penalizes long outputs when the answer is correct but encourages longer outputs for incorrect ones to allow more exploration. However, these approaches rely solely on correctness and largely ignore the underlying problem difficulty, which limits their ability to achieve optimal efficiency improvements. Ideally, from a human perspective, computational resources should be allocated adaptively, i.e., spending more effort on complex questions and less on simpler ones. Recent studies further suggest that difficult questions often benefit from longer chains of thought to maintain high performance, whereas simple problems can be solved with much shorter outputs or even without explicit reasoning (Liu et al., 2025b; Ma et al., 2025; Jiang et al., 2025; Fang et al., 2025). In the medical domain, pioneering work (Zhang et al., 2025) proposes decomposing the reasoning process into thinking and non-thinking modes based on the estimated difficulty of each question. However, their method requires human annotations to label questions as easy or hard, followed by training a separate classifier, which introduces manual effort, potential labeling bias, and scalability challenges.

Furthermore, recent studies highlight an inherent trade-off between performance and inference cost in hybrid/adaptive thinking models. For instance, Qwen3's hybrid reasoning model exhibits performance degradation across multiple tasks compared to separately trained thinking and non-thinking models, with the decline being particularly pronounced for the non-thinking component (Team, 2025). This limitation has led to a reversion to a dual-model strategy, employing separate models for reasoning-intensive and non-reasoning tasks (The Register, 2025). Similarly, while DeepSeekV3.1 (DeepSeek-AI, 2024) and GPT-5 also adopt hybrid reasoning architectures, they rely on additional large training datasets to mitigate performance loss. Nevertheless, user feedback suggests that hybrid models still deliver a less satisfactory experience compared to the dual-model paradigm. Thus, under comparable training conditions, balancing performance and inference cost in one model remains challenging, highlighting the need for more effective optimization strategies that reduce cost without compromising performance.

Motivated by the above observations, we propose AdaThink-Med, the first end-to-end framework designed to improve adaptive thinking capabilities in medical language models by explicitly incorporating problem difficulty. The central idea of AdaThink-Med is to enable the model to generate outputs of varying lengths depending on the difficulty of the input question, producing concise answers for simple cases and long reasoning chains for complex ones. During training, for each question, we generate multiple outputs and evaluate both their accuracy and uncertainty to derive a comprehensive measure of difficulty. This difficulty-aware signal is then incorporated into the reinforcement learning process, guiding the model to generate shorter outputs for low-difficulty with high-confidence right answer. At inference time, AdaThink-Med automatically estimates the difficulty of incoming questions and generates outputs with an appropriate amount of reasoning.

We conduct comprehensive experiments across both rule-based and model-based RL paradigms, and conduct the evaluation on six public medical reasoning benchmarks spanning various difficulty levels. Results show that AdaThink-Med consistently achieves better efficiency compared to existing medical language models and length-based output control methods. The contributions of this work are fourfold: (1) We present AdaThink-Med, the first end-to-end adaptive thinking framework for the medical domain, capable of autonomously switching between non-thinking and thinking modes during inference; (2) We propose a specific entropy-based difficulty estimator and plug it into a difficulty-aware length reward inside reinforcement fine-tuning, targeting medical reasoning models, and show emergent non-thinking vs. thinking modes; (3) We demonstrate the utility of AdaThink-Med in dataset selection, enabling efficient extraction of high-quality subsets from large-scale datasets; (4) We achieve state-of-the-art performance on multiple benchmarks, balancing accuracy and computational cost.

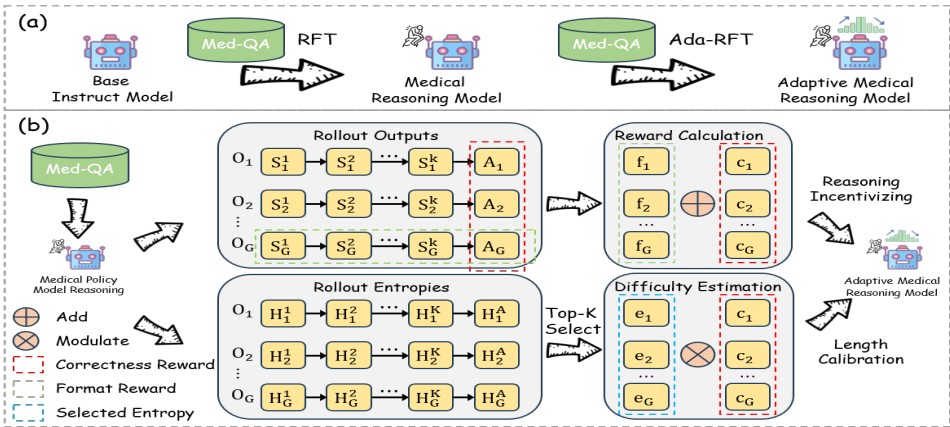

Figure 1: Overview of AdaThink-Med. (a) A medical reasoning model is first trained via RFT on medical QAs, then adaptively optimized to enhance reasoning efficiency; (b) entropy is used as an uncertainty measure to estimate problem difficulty prior to length calibration.

## 2 RELATED WORKS

**Efficient Reasoning for LLMs with Reinforcement Learning.** Efficient reasoning aims to reduce unnecessary computational cost while preserving accuracy, thereby mitigating the "overthinking phenomenon" that arises from excessively long Chain-of-Thought (CoT) sequences (Sui et al., 2025). Numerous recent studies aim to enhance the efficiency of reasoning models by controlling output length during reinforcement learning. Specifically, Kimi1.5 (Team et al., 2025) and DAPO (Yu et al., 2025) introduce a length penalty across all samples to discourage unnecessarily long responses. ShortBetter (Yi et al., 2025) proposes that the optimal output length should be the minimum length among correct responses and constrains the reasoning model accordingly. However, as discussed in Section 4.1, such greedy length-calibration strategies are prone to reward hacking and can lead to significant performance degradation, as they neither incentivize the model to extend reasoning when outputs are incorrect for hard problems nor account for variations in problem difficulty. CosFn (Yeo et al., 2025) attempts to alleviate performance loss by encouraging longer reasoning paths for incorrect responses, yet it still neglects to model problem difficulty explicitly. DAST (Shen et al., 2025) introduces problem difficulty adaptation but does not address the inherent uncertainty in estimating difficulty. Moreover, DAST employs offline training, which cannot adapt to the dynamic evolution of the model's reasoning capability. In contrast, our method uses entropy as a guidance signal for difficulty estimation and trains the reasoning model online, enabling adaptive thinking that evolves with the model's changing abilities.

**Entropy Guidance in LLM Post-Training**. In recent years, a growing body of research has investigated the use of entropy as a measure of uncertainty or difficulty to steer reasoning processes in large language models. For instance, (Chen et al., 2025b) incorporates semantic entropy as sample weighting in GRPO, observing that higher entropy often corresponds to harder problems demanding longer reasoning chains. (Xu et al., 2025) employs semantic entropy to dynamically terminate multi-step reasoning once uncertainty falls below a predefined threshold. (Li et al., 2025) leverages step-level entropy to prune components of chain-of-thought reasoning and trains models via GRPO to produce compressed reasoning traces. (Zhu et al., 2025) explicitly models the accuracy–length trade-off in terms of entropy and partitions the reasoning process into compression and exploration phases according to entropy dynamics. In contrast, AdaThink-Med utilizes entropy as a correctness-guided signal for accurate problem difficulty estimation, applying a threshold to distinguish between easy and hard cases for separate optimization. Furthermore, it represents the first application of entropy-guided reasoning to medical reinforcement learning, with the additional effect of enabling selective dataset use.

## 3 METHOD

In this section, we first present the necessary preliminaries. Then, more details of the proposed AdaThink-Med method for efficient reasoning during RL are presented.

**Problem Formulation.** Efficient reasoning aims to optimize a reasoning model $\pi_\theta$ parameterized by $\theta$ to generate high-quality outputs with reduced inference cost. Given a dataset $\mathcal{D}$, where each sample consists of an input prompt $\mathbf{x}$ and the corresponding ground-truth answer $\mathbf{y}^*$, the model produces an output $\mathbf{y} \sim \pi_\theta(\cdot|\mathbf{x})$. A reward function $R(\mathbf{y}, \mathbf{y}^*)$ is defined to evaluate the correctness of the generated output $\mathbf{y}$. To ensure stable optimization and prevent overfitting, a reference model $\pi_{\text{ref}}$ is introduced to regularize the policy update via a KL-divergence penalty term. The optimization objective for efficient reasoning is defined as:

$$\max_\theta \; \mathbb{E}_{\mathbf{x} \sim \mathcal{D}, \mathbf{y} \sim \pi_\theta(\cdot|\mathbf{x})} \Big[ R(\mathbf{y}, \mathbf{y}^*) \; - \; \lambda \cdot \mathcal{C}(\pi_\theta, \mathbf{x}) \; - \; \beta \cdot D_{\text{KL}}\big(\pi_\theta(\mathbf{y}|\mathbf{x}) \,\|\, \pi_{\text{ref}}(\mathbf{y}|\mathbf{x})\big) \Big], \quad (1)$$

where $\mathcal{C}(\pi_\theta, \mathbf{x})$ denotes the computational cost of generating $\mathbf{y}$ for the input $\mathbf{x}$, $\lambda$ controls the trade-off between reward and computational efficiency, and $\beta$ controls the influence of the KL-divergence regularization term. The reference model $\pi_{\text{ref}}$ corresponds to the model state prior to efficient reasoning optimization. We use GRPO as our reinforcement learning method and the details can be found at Appendix B.

### 3.1 ADATHINK-MED

To build an efficient LLM, adaptive thinking enables models to think more on complex questions and less on simple ones, thereby reducing total token consumption to meet the target in equation 1. A key challenge in this endeavor is accurately distinguishing between easy and hard questions—that is, estimating problem difficulty appropriately. An intuitive approach is to measure problem difficulty via sampling accuracy (Team et al., 2025; Tong et al., 2024), defined as the proportion of correct outputs among a set of candidate responses. This can be formulated as:

$$\mathcal{D}_{acc} = 1 - \frac{1}{G} \sum_{i=1}^{G} \mathbb{I}[o_i = y^*], \quad (2)$$

where $G$ denotes the number of reasoning rollouts per question $\mathbf{x}$, $o_i \in \{o_1, o_2, \ldots, o_G\}$ represents the model's output in the $i$-th rollout, $y^*$ is the ground-truth answer, and $\mathbb{I}[\cdot]$ is the indicator function. However, this sample-level difficulty estimation requires a large number of output samples ($G$) to yield a stable result. Moreover, it overlooks the inherent uncertainty in each generated output. Since LLM generations exhibit considerable variability, this approach treats all correct answers equally without accounting for underlying uncertainty.

To address this limitation, we propose that incorporating output-level uncertainty estimation into the difficulty assessment framework is essential. Inspired by recent studies on uncertainty quantification in LLM reasoning via entropy analysis (Wang et al., 2025; Cui et al., 2025; Cheng et al., 2025), we introduce AdaThink-Med, an Uncertainty-Guided difficulty-aware length calibration algorithm. The method is structured into four components:

**1) Output Uncertainty Calculation.** Following (Wang et al., 2025), we define the token-level entropy at decoding step $t$ as:

$$H_t := -\sum_{j=1}^{V} p_{t,j} \log p_{t,j}, \quad \text{where } p_t = (p_{t,1}, \cdots, p_{t,V}) = \pi_\theta(\cdot \mid \mathbf{x}, s_{<t}) = \text{Softmax}\left(\frac{z_t}{T}\right). \quad (3)$$

In this formulation, $s_{<t} = (s_1, s_2, \ldots, s_{t-1})$ denotes the sequence of tokens generated prior to time step $t$. The vocabulary size is $V$, and the vector $z_t \in \mathbb{R}^V$ represents the unnormalized model logits before the softmax transformation. These logits are scaled by a temperature parameter $T \in \mathbb{R}$ and converted into a probability distribution $p_t \in \mathbb{R}^V$ over the vocabulary. When using an off-policy setup, the generation of token sequences may rely on a separate rollout policy $\pi_\phi$, while training still optimizes a target policy $\pi_\theta$ with distinct parameters ($\phi \neq \theta$). In such cases, the entropy $H_t$ is evaluated using the distribution induced by the training policy $\pi_\theta$, as specified in equation 3, in order to quantify the uncertainty of $\pi_\theta$ conditioned on the observed prefix $s_{<t}$. To quantify the overall

uncertainty of the output sequence, we follow the approach proposed in (Wang et al., 2025), which demonstrates that the mean entropy of the top-$K$ most uncertain tokens (e.g., top 20%) effectively represents the entropy of the entire trajectory.

Accordingly, we define the output-level uncertainty as the average entropy over the top-$K$ tokens with the highest token-level entropy values. To ensure comparability across samples within a batch, we apply min-max normalization to the resulting trajectory-level uncertainties. Specifically, for a batch of $\mathcal{B}$ sequences, each uncertainty score $\mathcal{H}_i$ is normalized:

$$\mathcal{H}_i := \frac{1}{K} \sum_{t \in \mathcal{T}_K} H_t, \widetilde{\mathcal{H}}_i := \frac{\mathcal{H}_i - \min_j \mathcal{H}_j}{\max_j \mathcal{H}_j - \min_j \mathcal{H}_j}, \quad \text{for } i = 1, \dots, |\mathcal{B}| \times G. \quad (4)$$

where $\mathcal{T}_K$ is the set of indices corresponding to the top-$K$ highest values of token entropy $\{H_t\}_{t=1}^{|o_i|}$ in the sequence of length $|o_i|$.

**2) Uncertainty-guided Difficulty Estimation.** We quantify problem difficulty by discounting correctness using the trajectory-level uncertainty. Given $G$ independent responses to a question with ground-truth answer $y^*$, let $o_i$ denote the $i$-th predicted answer and $\widetilde{\mathcal{H}}_i \in [0, 1]$ be its batch-wise min–max normalized trajectory entropy. The difficulty score $\mathcal{D}_q$ is defined as:

$$\mathcal{D}_q = 1 - \frac{1}{G} \sum_{i=1}^{G} \alpha(1 - \widetilde{\mathcal{H}}_i)\mathbb{I}[\, o_i = y^*\,] + (1 - \alpha)\mathbb{I}[\, o_i = y^*\,]. \quad (5)$$

This formulation assigns higher difficulty to questions where correct answers are accompanied by greater uncertainty, or where correctness is sparse across responses.

**3) Length Reward Calibration.** To distinguish between easy and difficult questions, we introduce a dynamic difficulty threshold $\theta_{\mathcal{B}}$, computed as the $\tau$-quantile over the difficulty scores of all questions in the batch $\mathcal{B}$ as $\theta_{\mathcal{B}} = \text{Quantile}(\{\mathcal{D}_q \mid q = 1, \dots, |\mathcal{B}|\}, \tau)$, where $\tau$ is a hyperparameter controlling the easy–hard split. The length-based reward $\mathcal{R}_{\text{len}}^i$ is defined as:

$$\mathcal{R}_{\text{len}}^i = \begin{cases} \max\left(0, 1 - \frac{L_i}{\bar{L}_s} \cdot \rho_i\right) & \text{if } \mathcal{D}_q < \theta_{\mathcal{B}} \text{ and } \mathbb{I}[\, o_i = y^*\,], \\ \min\left(1, \frac{L_i}{\bar{L}_h} \cdot \rho_i - 1\right) & \text{if } \mathcal{D}_q > \theta_{\mathcal{B}} \text{ and } \mathbb{I}[\, o_i \neq y^*\,], \\ 0 & \text{otherwise.} \end{cases} \quad (6)$$

Here, $\bar{L}_s$ and $\bar{L}_h$ denote the average output lengths for simple and hard questions, respectively, and $\rho_i = 0.5 + 0.5 \times (1 - \widetilde{\mathcal{H}}_i)$ is a scaling factor. The second row represents performance compensation, which penalizes shorter outputs when accuracy declines and thereby encourages the model to extend its reasoning length on difficult problems with wrong outputs. The motivation for this design is further elaborated in the **Length Reward Hack** part (Section 4.1).

The final reward for each output $o_i$ combines accuracy, format adherence, and length calibration:

$$\mathcal{R}_i = \mathcal{R}_{\text{acc}}^i + 0.5 \cdot \mathcal{R}_{\text{format}}^i + 0.5 \cdot \mathcal{R}_{\text{len}}^i. \quad (7)$$

**4) Momentum-based smoothing & staged training.** To enhance training stability, we apply exponential moving average (EMA) updates to key batch-level statistics. Specifically, the difficulty threshold $\theta_{\mathcal{B}}$ and average lengths $\bar{L}_s$, $\bar{L}_h$ are updated as $\theta_{\mathcal{B}} \leftarrow \alpha \cdot \theta_{\mathcal{B}}^{\text{cur}} + (1 - \alpha) \cdot \theta_{\mathcal{B}}^{\text{old}}$, where $\alpha$ is the momentum coefficient (e.g., $\alpha = 0.1$). Similar updates apply for $\bar{L}_s$ and $\bar{L}_h$. Besides, as illustrated in Fig. 1, we first train an LLM without length calibration to obtain a medical reasoning model, and then apply reinforcement learning with length calibration to enforce adaptive medical reasoning.

## 4 EXPERIMENTS

**Training Datasets and Evaluation Benchmarks.** We use the AlphaMed19k (Liu et al., 2025a) dataset as our primary training corpus. AlphaMed19k integrates the official training splits (Huang et al., 2025) of two public multiple-choice medical QA benchmarks, MedQA (Jin et al., 2021) and MedMCQA (Pal et al., 2022). MedQA contains expert-level clinical questions drawn from the

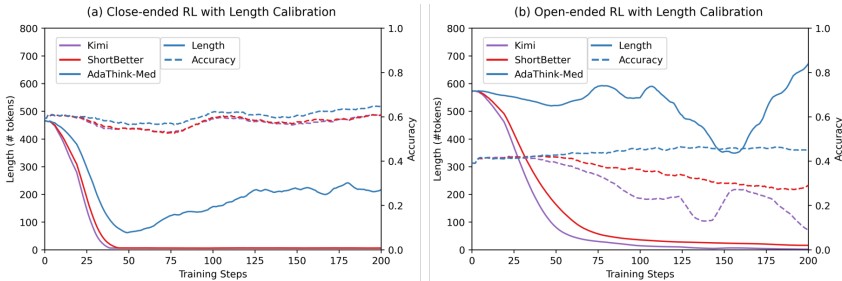

Figure 2: Length reward hack during RL with greedy length calibration. (a) RL with a close-ended setup; (b) RL with an open-ended setup.

USMLE, while MedMCQA covers factoid and reasoning questions from Indian medical entrance examinations. For out-of-domain evaluation we use PubMedQA (Jin et al., 2019), the medical subsets of MMLU-Pro (Wang et al., 2024) (MMLU-ProM), the medical subsets of GPQA (Rein et al., 2024) (GPQA-M), and the recent clinical benchmark MedXpertQA (Zuo et al., 2025). Additional dataset details are provided in the Appendix B.4.

**Baseline & Implementation Details.** We compare AdaThink-Med with baseline models from three categories: general zero-shot LLMs (Qwen-2.5-Instruct-7B, Llama-3.1-Instruct-8B), medical tuned LLMs (MedLlama3, MMed, Med42 (Christophe et al., 2024), OpenBioLLM (Pal & Sankarasubbu, 2024), UltraMedical (Zhang et al., 2024), HuatuoGPT-o1 (Chen et al., 2024b), m1 (Huang et al., 2025)), and length-calibration methods (Kimi1.5 (Team et al., 2025), ShortBetter (Yi et al., 2025), DAST (Shen et al., 2025), CosFn (Yeo et al., 2025)). More details of baselines and implementation are summarized in Appendix B.2 and Appendix B.3, respectively.

**Evaluation Metrics.** We report accuracy and output length to assess correctness against ground truth and token usage, respectively. To capture the trade-off between computational cost and performance, we adopt the **Accuracy-Efficiency Score (AES)** (Luo et al., 2025), defined as:

$$\text{AES} = \begin{cases} \alpha \cdot \Delta\text{Length} + \beta \cdot |\Delta\text{Acc}|, & \text{if } \Delta\text{Acc} \geq 0, \\ \alpha \cdot \Delta\text{Length} - \gamma \cdot |\Delta\text{Acc}|, & \text{if } \Delta\text{Acc} < 0. \end{cases}$$

Here, $\Delta\text{Length} = \frac{\text{Length}_{\text{baseline}} - \text{Length}_{\text{model}}}{\text{Length}_{\text{baseline}}}$ is the relative reduction in output length and $\Delta\text{Acc} = \frac{\text{Acc}_{\text{model}} - \text{Acc}_{\text{baseline}}}{\text{Acc}_{\text{baseline}}}$ is the relative change in accuracy. We set $\alpha = 1$, $\beta = 3$, and $\gamma = 5$ by default. The asymmetric design places a larger penalty on accuracy drops than the reward for improvements, which aligns with practical preferences.

## 4.1 RESULTS

**Performance Comparison with General and Medical Reasoning LLMs.** Reducing inference cost while improving reasoning performance is essential in real-workd application of medical LLMs . We first investigate the extent to which models trained with AdaThink-Med enhance inference efficiency compared with both general and medical reasoning LLMs. As shown in Table 1, AdaThink-Med achieves consistent efficiency gains across six benchmarks. Specifically, AdaThink-Med-LLaMA and AdaThink-Med-Qwen outperform their respective baselines (GRPO) with average efficiency scores of 0.75 and 0.93. Notably, AdaThink-Med-Qwen improves accuracy by 0.25% while reducing output length by 78.67%, and AdaThink-Med-LLaMA compresses output length by 84.39% with only a 1.13% accuracy drop. Although some medical LLM baselines achieve higher accuracy, they require substantially greater inference cost and more training data. In contrast, AdaThink-Med is trained only on 19k samples without any distilled CoT data, yet achieves state-of-the-art efficiency and shows strong scaling potential. Moreover, on easier benchmarks such as MedQA and PubMedQA, AdaThink-Med generates significantly fewer output tokens compared with harder benchmarks such as MedMCQA, MMLU-ProM, and GPQA-M, highlighting the effectiveness of its adaptive thinking strategy. Overall, AdaThink-Med provides highly efficient inference across diverse benchmarks and demonstrates strong potential for efficient medical reasoning models.

Table 1: Performance of models on in-domain (*) and out-of-domain medical QA benchmarks. The best results are shown in **bold**, and the second-best are underlined. + indicates reasoning ability obtained with a Chain-of-Thought prompt during inference; † denotes reasoning ability learned through supervised fine-tuning with long CoT data; ◇ represents reasoning ability acquired via reinforcement learning; and ‡ refers to training with additional datasets beyond MedQA and MedMCQA. For AES, baseline model is LLama-3.1-8B-Instruct+.

| Model | MedQA* | | MedMCQA* | | PubMedQA | | MMLU-ProM | | GPQA-M | | MedXpert | | Avg | | AES |
|---|---|---|---|---|---|---|---|---|---|---|---|---|---|---|---|
| | Acc. | Len. | Acc. | Len. | Acc. | Len. | Acc. | Len. | Acc. | Len. | Acc. | Len. | Acc. | Len. | |
| *Llama backbone* | | | | | | | | | | | | | | | |
| Llama-3.1-8B-Instruct+ | 68.18 | 461 | 57.32 | 309 | 78.20 | 296 | 60.06 | 457 | 45.64 | 485 | 16.08 | 545 | 54.25 | 425 | – |
| Llama-3.1-8B-Instruct | 54.75 | 302 | 54.69 | 169 | 77.00 | 113 | 57.00 | 256 | 35.64 | 345 | 14.48 | 342 | 48.93 | 254 | -0.09 |
| MedLlama3-8B-v1+ | 50.90 | 187 | 41.47 | 153 | 46.00 | 133 | 30.68 | 234 | 32.05 | 265 | 12.20 | 310 | 35.55 | 214 | -1.23 |
| MedLlama3-8B-v2+ | 60.56 | 271 | 54.53 | 254 | 73.40 | 216 | 54.26 | 323 | 43.84 | 378 | 13.91 | 315 | 50.08 | 293 | -0.07 |
| MMed-8B†‡+ | 49.01 | 888 | 43.82 | 843 | 59.40 | 832 | 34.65 | 856 | 31.79 | 831 | 13.06 | 938 | 38.62 | 865 | -2.47 |
| Med42-8B‡+ | 57.97 | 343 | 55.53 | 276 | 70.80 | 204 | 53.28 | 357 | 43.07 | 359 | 13.26 | 402 | 48.99 | 324 | -0.25 |
| OpenBioLLM-8B†‡◇ | 50.51 | 183 | 44.84 | 160 | 55.40 | 141 | 38.82 | 153 | 31.28 | 167 | 12.16 | 207 | 38.84 | 169 | -0.82 |
| UltraMedical-8B-3†‡◇ | 70.30 | 402 | 59.16 | 364 | 76.80 | 282 | 59.93 | 524 | 46.92 | 490 | 15.91 | 630 | 54.84 | 449 | -0.02 |
| UltraMedical-8B-3.1†‡◇ | 74.46 | 486 | 62.65 | 450 | 78.50 | 392 | 62.93 | 603 | 46.92 | 588 | 16.89 | 710 | 57.06 | 538 | -0.11 |
| HuatuoGPT-o1-8B†‡◇ | **76.35** | 568 | **62.82** | 445 | **79.80** | 445 | 63.71 | 521 | **54.35** | 585 | 17.06 | 601 | **59.02** | 527 | +0.02 |
| GRPO-LLama◇ | 72.50 | 438 | 61.60 | 299 | 78.50 | 290 | 63.51 | 442 | 46.92 | 461 | 17.26 | 529 | 56.72 | 410 | +0.17 |
| **AdaThink-Med-LLama** | 67.00 | **36** | 60.41 | **73** | 78.40 | **54** | 63.58 | **85** | 47.94 | **100** | 16.20 | **35** | 55.59 | **64** | **+0.92** |
| *Qwen backbone* | | | | | | | | | | | | | | | |
| Qwen2.5-7B-Instruct+ | 63.86 | 490 | 56.68 | 359 | 73.10 | 386 | 62.28 | 523 | 46.41 | 562 | 12.28 | 585 | 52.44 | 484 | - |
| Qwen2.5-7B-Instruct | 54.28 | 313 | 53.43 | 196 | 72.70 | 129 | 56.67 | 292 | 38.71 | 448 | 12.44 | 291 | 48.04 | 278 | -3.73 |
| m1-7B†‡ | 75.01 | 2161 | 62.32 | 1749 | 74.20 | 1080 | **68.07** | 2564 | 51.53 | 3887 | **18.28** | 3265 | 58.24 | 2451 | -4.54 |
| GRPO-Qwen◇ | 67.55 | 504 | 59.52 | 378 | 72.30 | 396 | 66.12 | 535 | 48.46 | 575 | 14.48 | 592 | 54.74 | 497 | +0.11 |
| **AdaThink-Med-Qwen** | 68.34 | 79 | 58.74 | 109 | 73.50 | 62 | 66.45 | 121 | 48.72 | 175 | 14.16 | 88 | 54.99 | 106 | +0.93 |

**Length Reward Hack & Motivation of AdaThink-Med's Performance Compensation.** We conduct pilot experiments to investigate greedy output length reduction strategies that do not explicitly incentivize incorrect responses. Two representative approaches are Kimi1.5 (Team et al., 2025), which enforces uniform length reduction across all samples, and ShortBetter (Yi et al., 2025), which drives responses toward the minimal length among correct samples. As shown in Fig. 2(a), the absence of such compensation causes both Kimi1.5 and ShortBetter to collapse in terms of output length and accuracy. We initially suspect that this collapse primarily arises in close-ended tasks, where the presence of candidate answers allows the model to adopt a trivial strategy of directly outputting the final answer without any reasoning. To avoid this shortcut, we adopt an open-ended RL setting following HuatuoGPT-o1 (Chen et al., 2024b), where the model must reason and answer using its world knowledge without being provided candidate options (Rui et al., 2025a). Implementation details are given in the Appendix. As shown in Fig. 2(b), even in the open-ended setting, greedy length calibration leads to collapse, with both accuracy and output length dropping sharply and failing to recover. Moreover, we found that the collapsed model no longer produces meaningful answers or follows instructions. We attribute this phenomenon to a form of length reward hack: in its search for shorter responses, the model converges to a minimal-length strategy that maintains apparent accuracy while maximizing length-based rewards. This shortcut undermines the emergence of diverse reasoning patterns and degrades instruction following. To address this issue, we introduce a performance compensation mechanism that penalizes shorter outputs when accuracy degrades. This encourages the model to achieve a more balanced trade-off between performance and efficiency, ultimately leading to stable training dynamics as shown in Fig. 2.

**Comparison with Length-calibration Methods.** As shown in Table 2, AdaThink-Med delivers marked efficiency gains over existing length-calibration baselines. It reduces the average reasoning length to 64 tokens on Llama and 106 tokens on Qwen. These lengths are 2.0× and 2.8× shorter than those produced by Kimi1.5, and 2.6× and 3.2× shorter than those from ShortBetter, respectively, while achieving comparable accuracy (Llama: 55.59% vs. 55.94% for ShortBetter; Qwen: 54.99% vs. 54.28% for ShortBetter). Additionally, AdaThink-Med achieves the highest AES scores on both model backbones, with 0.92 on Llama and 0.93 on Qwen, respectively. This corresponds to an improvement of +0.14 over the next best method (Kimi1.5) on Llama and +0.38 on Qwen. These improvements stem from the model's adaptive thinking capability: AdaThink-Med dynamically allocates reasoning length, maintaining brevity on easy questions while employing extended reasoning chains on challenging or initially incorrect cases through its accuracy aware compensation mechanism. This approach effectively reduces computational cost without sacrificing reliability.

Table 2: Performance of length calibration methods on in-domain (*) and out-of-domain medical QA benchmarks. The best results are shown in **bold**, and the second-best are underlined. For AES, baseline model is LLama-3.1-8B-Instruct[+] and Qwen-2.5-7B-Instruct[+], respectively.

| Model | MedQA* | | MedMCQA* | | PubMedQA | | MMLU-ProM | | GPQA-M | | MedXpert | | Avg | | AES |
|---|---|---|---|---|---|---|---|---|---|---|---|---|---|---|---|
| | Acc. | Len. | Acc. | Len. | Acc. | Len. | Acc. | Len. | Acc. | Len. | Acc. | Len. | Acc. | Len. | |
| *Llama backbone* | | | | | | | | | | | | | | | |
| Llama-3.1-8B-Instruct+ | 68.18 | 461 | 57.32 | 309 | 78.20 | 296 | 60.06 | 457 | 45.64 | 485 | 16.08 | 545 | 54.25 | 425 | – |
| Llama-3.1-8B-Instruct | 54.75 | 302 | 54.69 | 169 | 77.00 | 113 | 57.00 | 256 | 35.64 | 345 | 14.48 | 342 | 48.93 | 254 | -0.09 |
| Llama-3.1-8B-Instruct-GRPO | 72.50 | 438 | 61.60 | 299 | 78.50 | 290 | 63.51 | 442 | 46.92 | 461 | 17.26 | 529 | 56.72 | 410 | +0.17 |
| Kimi1.5 (Team et al., 2025) | 70.62 | 145 | 60.93 | 100 | 77.50 | 92 | 63.71 | 164 | 45.38 | 118 | 15.75 | 131 | 55.65 | 125 | +0.78 |
| CosFn (Yeo et al., 2025) | 37.94 | 1007 | 49.29 | 919 | 75.10 | 885 | 23.32 | 998 | 32.82 | 1012 | 10.12 | 1020 | 38.10 | 974 | -2.78 |
| DAST (Shen et al., 2025) | 60.64 | 30 | 56.9 | 34 | 76.00 | 39 | 59.48 | 71 | 47.69 | 53 | 15.35 | 82 | 52.71 | 51 | +0.74 |
| ShortBetter (Yi et al., 2025) | 71.48 | 175 | 60.62 | 134 | 78.60 | 118 | 62.99 | 188 | 45.89 | 219 | 16.08 | 171 | 55.94 | 167 | +0.70 |
| **AdaThink-Med-LLama** | 67.00 | 36 | 60.41 | 73 | 78.40 | 54 | 63.58 | 85 | 47.94 | 100 | 16.20 | 35 | 55.59 | 64 | **+0.92** |
| *Qwen backbone* | | | | | | | | | | | | | | | |
| Qwen2.5-7B-Instruct+ | 63.86 | 490 | 56.68 | 359 | 73.10 | 386 | 62.28 | 523 | 46.41 | 562 | 12.28 | 585 | 52.44 | 484 | – |
| Qwen2.5-7B-Instruct | 54.28 | 313 | 53.43 | 196 | 72.70 | 129 | 56.67 | 292 | 38.71 | 448 | 12.44 | 291 | 48.04 | 278 | +0.01 |
| Qwen2.5-7B-Instruct-GRPO | 67.55 | 504 | 59.52 | 378 | 72.30 | 396 | 66.12 | 535 | 48.46 | 575 | 14.48 | 592 | 54.74 | 497 | +0.11 |
| Kimi1.5 (Team et al., 2025) | 69.99 | 285 | 58.95 | 225 | 75.00 | 236 | 66.71 | 338 | 46.92 | 376 | 14.89 | 330 | 55.41 | 298 | +0.55 |
| CosFn (Yeo et al., 2025) | 37.94 | 1007 | 49.29 | 919 | 75.10 | 885 | 23.32 | 998 | 32.82 | 1012 | 10.12 | 1020 | 38.10 | 974 | -2.38 |
| DAST (Shen et al., 2025) | 65.90 | 260 | 59.67 | 327 | 69.70 | 153 | 65.86 | 205 | 50.51 | 466 | 15.59 | 288 | 54.54 | 283 | +0.54 |
| ShortBetter (Yi et al., 2025) | 67.94 | 333 | 58.90 | 250 | 71.70 | 260 | 65.86 | 376 | 46.92 | 405 | 14.36 | 387 | 54.28 | 335 | +0.41 |
| **AdaThink-Med-Qwen** | 68.34 | 79 | 58.74 | 109 | 73.50 | 62 | 66.45 | 121 | 48.72 | 175 | 14.16 | 88 | 54.99 | 106 | **+0.93** |

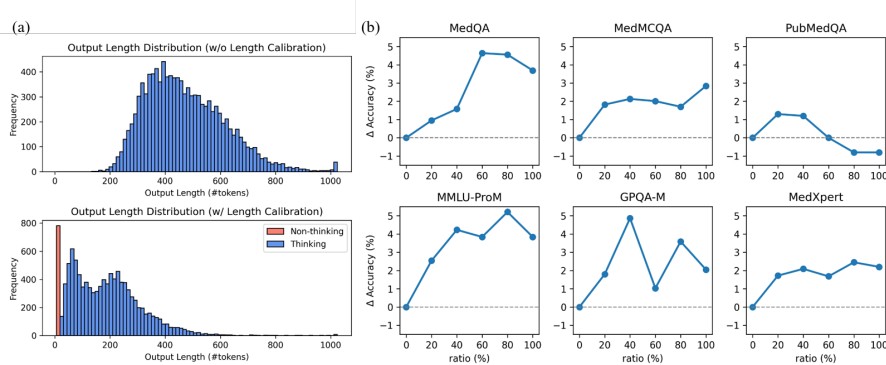

Figure 3: (a) Output length distribution w and w/o length calibration. (b) Dataset selection results.

**Emergence of Reasoning Mode Divergence.** As illustrated in Fig. 3(a), models trained with AdaThink-Med exhibit a notable divergence in reasoning modes, spontaneously developing two distinct behaviors: a *non-thinking* mode that outputs answers directly without intermediate reasoning, and a *thinking* mode characterized by concise yet essential reasoning steps. Unlike prior work (Zhang et al., 2025), which relies on manually annotated data, classifier-based pipelines, and enforced formatting to induce such separation, our results show for the first time in medical QA that this reasoning dichotomy can emerge end-to-end, without any additional supervision.

**Dataset Selection.** Dataset selection (Albalak et al., 2024) refers to choosing a high-quality subset on which training achieves comparable performance without the full set. AdaThink-Med model tends to produce longer outputs on harder problems and shorter outputs on easier ones, identifying high-value data for reinforcement learning. Concretely, we run the trained model (e.g., AdaThink-Med-Qwen) on the original Alphamed19k dataset to obtain the output length distribution, then partition the samples into short and long groups using the global median output length as the threshold. We next perform stratified sampling to keep the proportions of easy and hard questions balanced, and construct subsets that include both groups at 20%, 40%, 60%, and 80% of the full set to study the optimal compression ratio. As shown in Fig. 3(b) and detailed in the Appendix, a 40% compression ratio yields performance that exceeds training on the full dataset, with 2.10% gains on PubMedQA and 2.82% on GPQA-M. Even at 20%, the model preserves 98.8% of the full-set performance, which supports the effectiveness of AdaThink-Med for dataset selection.

**Impact of Problem Difficulty $\tau$ and Uncertainty Weight $\alpha$.** Since the parameter $\tau$ coarsely determines the overall compression ratio of the response length, we first fix the difficulty metric by setting the weight $\alpha$ for the uncertainty measure to 0.5, in order to investigate the impact of the

Table 3: Ablation study on $\tau$ and $\alpha$ across in-domain and out-of-domain datasets. We rewrite the result of $\tau = 0.7, \alpha = 0.5$ for clear comparison.

| $\tau$ | $\alpha$ | MedQA | | MedMCQA | | PubMedQA | | MMLU-ProM | | GPQA-M | | MedXpert | | Avg | | AES |
|---|---|---|---|---|---|---|---|---|---|---|---|---|---|---|---|---|
| | | Acc. | Len. | Acc. | Len. | Acc. | Len. | Acc. | Len. | Acc. | Len. | Acc. | Len. | Acc. | Len. | |
| 0.3 | 0.5 | 68.19 | 391 | 58.76 | 307 | 73.90 | 311 | 68.08 | 434 | 50 | 483 | 14.37 | 457 | 55.55 | 397 | 0.36 |
| 0.5 | 0.5 | 67.32 | 105 | 59.12 | 170 | 72.80 | 84 | 67.17 | 196 | 47.69 | 245 | 15.02 | 116 | 54.85 | 153 | 0.82 |
| 0.7 | 0.5 | 68.34 | 79 | 58.74 | 109 | 73.50 | 62 | 66.45 | 121 | 48.72 | 175 | 14.16 | 88 | 54.99 | 106 | 0.93 |
| 0.9 | 0.5 | 62.53 | 7 | 58.50 | 13 | 74.00 | 6 | 64.30 | 11 | 46.67 | 12 | 14.49 | 8 | 53.41 | 10 | 1.04 |
| 0.7 | 0.1 | 66.06 | 121 | 59.02 | 153 | 74.30 | 116 | 66.19 | 195 | 50.51 | 258 | 15.39 | 142 | 55.25 | 164 | 0.82 |
| 0.7 | 0.5 | 68.34 | 79 | 58.74 | 109 | 73.50 | 62 | 66.45 | 121 | 48.72 | 175 | 14.16 | 88 | 54.99 | 106 | 0.93 |
| 0.7 | 0.9 | 63.47 | 44 | 58.12 | 216 | 73.60 | 10 | 63.19 | 50 | 46.92 | 69 | 14.08 | 30 | 53.23 | 70 | 0.90 |

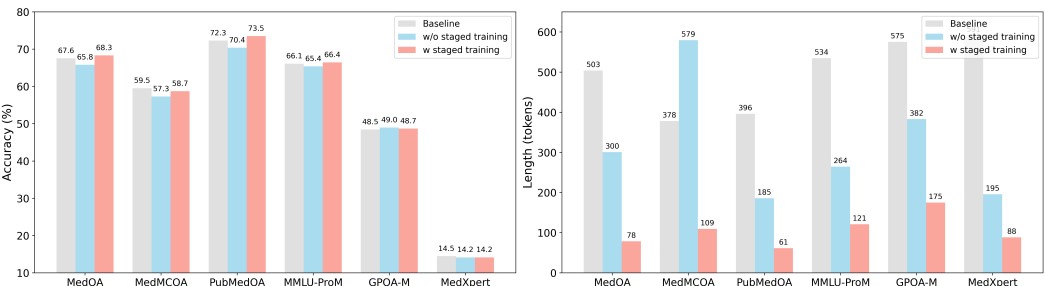

Figure 4: Performance comparison w and w/o staged adaptive training.

difficulty threshold $\tau$. A grid-search over $\{0.3, 0.5, 0.7, 0.9\}$ is conducted for the optimal $\tau$. As shown in Table 3, when $\tau$ is set too high (0.9), the average length on the test set is substantially reduced along with a degradation in performance and length reward hack. Conversely, $\tau = 0.3$ produces longer reasoning chains across the dataset (redundant over-thinking), even though performance improves slightly. Next, we examine two values for $\alpha$, i.e., 0.1 and 0.9, while fixing $\tau$. Table 3 shows that $\alpha = 0.1$ (accuracy-focused) gives weaker performance (AES 0.82), while $\alpha = 0.9$ (uncertainty-focused) leads to only a slight drop (AES 0.90), highlighting the benefit of uncertainty-based difficulty for efficient reasoning. $\alpha = 0.5$ exhibits the best performance among the candidates. We believe an over-reliance on the uncertainty measure ($\alpha = 0.9$) may lead to a lag in performance compensation as its estimation precision increases. Overall, $\alpha = 0.5$ is identified as the optimal setting under the balanced $\tau = 0.7$.

**Effectiveness of Staged Adaptive Training.** In the initial phase of reinforcement learning, the model often exhibits low prediction accuracy and significant uncertainty, causing most questions to be perceived as hard. Emphasizing samples with incorrect outputs, our model inherently prioritizes accuracy improvement, leading to an imbalance between reducing overthinking and enhancing overall performance. To address this, we introduce a staged adaptive training strategy: the base model is first trained to convergence without length calibration, after which adaptive training is applied. As illustrated in Fig. 4, omitting the staged approach leads to limited length compression and noticeable performance degradation. In contrast, staged training maintains performance while achieving greater reduction in output length, demonstrating its effectiveness.

**Insufficient Reasoning in Medium-Difficulty Tasks Due to Token Depletion** Upon careful examination, we identified that for medium-difficulty tasks, the model struggled with balancing non-thinking and thinking, often relying on quick thinking that resulted in insufficient reasoning. In failure case (a), the model's output incorrectly suggests that global management might improve outcomes based on trends toward better disease-free survival, despite the lack of statistically significant differences in overall survival and disease-free survival between global management and separate management. This reflects a failure in properly interpreting the study results, where no significant differences in survival were found, and the disease-free survival advantage for SM was statistically significant. The model appears to overemphasize minor trends, neglecting the statistical evidence that showed global management did not provide superior results in the key outcomes. This oversight underscores the importance of distinguishing between statistical significance and potential clinical

**Question:** Synchronous liver metastases (SLM) occur in 20% of colorectal cancers (CRC). Resection of SLM and CLC can be undertaken at different centres (separate management, SM) or at the same centre (global management, GM). Retrospective study of SLM and CRC resections carried out during 01/2000 - 12/2006 by SM or GM, using a combined or delayed strategy. Morphologic characteristics and type of CRC and SLM resection were similar for the GM (n = 45) or SM (n = 66) groups. In patients with delayed liver resection (62 SM, 17 GM), chemotherapy prior to liver surgery was used in 92% and 38% of SM and GM patients (P<0.0001) and the median delay between procedures was 212 and 182 days, respectively (P = 0.04). First step of liver resection was more often performed during colorectal surgery in the GM group (62 vs. 6% for SM, P<0.0001) and the mean number of procedures (CRC+SLM) was lower (1.6 vs. 2.3, P = 0.003). Three-month mortality was 3% for GM and 0% for SM (n.s.). Overall survival rates were 67% and 51% for SM and GM at 3 years (n.s.), and 35 and 31% at 5 years (n.s.). Disease-free survival to 5 years was higher in SM patients (14% vs. 11%, P = 0.009).Using the above document as a reference, please review the following question and choose the option you find most appropriate. Colorectal cancer with synchronous liver metastases: does global management at the same centre improve results? Options: A: yes B: no C: maybe

**Ground Truth: B**

**Response:** Based on the information provided, the overall survival and disease-free survival rates did not significantly differ between the global management and separate management groups. However, the number of procedures was lower and chemotherapy use before liver surgery was more frequent in the global management group, which could imply better coordination and possibly better outcomes. Given this context, the best answer is:\\boxed{A}While the survival rates are not significantly different, the trend towards better disease-free survival suggests that global management might have advantages, even if not statistically significant for all measures. ✗

(a) Failure Case 1

**Question:** A 37-year-old woman with right lower extremity edema is evaluated because of the sudden onset of shortness of breath and pleuritic chest pain. A diagnosis of pulmonary embolism is made. Which of the following signs, if present on physical examination, would be the most specific indicator of pulmonary arterial hypertension in this patient? Options: A: Blue lips or skin B: Peripheral edema C: Rapid heart rate D: Constant chest pain E: Presence of an S3F: Wheezing or whistling sound when breathing G: Shortness of breath after slight activity H: P2 louder than A2I: Increased jugular venous pressure J: Coughing up blood. Please reason step by step, and put the final answer in \\boxed{}.

**Ground Truth: H**

**Response:** The most specific sign for pulmonary arterial hypertension in this scenario would be an S3 gallop heard on auscultation, as it suggests left ventricular dysfunction which can occur secondary to chronic pulmonary hypertension.\\boxed{E}. ✗

(b) Failure Case 2

Figure 5: Failure Cases Study

trends, particularly when the data does not support a definitive advantage for one management strategy over the other. Consequently, a more nuanced approach should have been employed to align the analysis with the study's findings. In failure case (b), the model incorrectly identifies the S3 gallop (E) as the most specific sign for pulmonary arterial hypertension. While an S3 sound can indicate left ventricular dysfunction, it is more commonly associated with volume overload or heart failure, not specifically PAH. In contrast, the sign of P2 louder than A2 (H) is far more specific to pulmonary arterial hypertension, as it directly reflects increased pressure in the pulmonary arteries. The model failed to consider this crucial auscultatory finding, which is the hallmark sign of PAH, particularly as the increased pulmonary pressure causes delayed and louder closure of the pulmonary valve (P2). Thus, the model did not appropriately prioritize the most specific sign for PAH.

## 5 CONCLUSION

In this paper, we propose **AdaThink-Med**, an Uncertainty-Guided adaptive thinking framework designed to improve the efficiency of medical large language models. By dynamically shaping output lengths based on problem difficulty, AdaThink-Med effectively reduces redundant reasoning while maintaining competitive accuracy. Furthermore, we propose a data selection strategy that identifies high-quality reasoning and non-reasoning samples, enabling efficient training with significantly fewer data. Extensive experiments across six medical reasoning benchmarks demonstrate that AdaThink-Med achieves state-of-the-art overall performance under the trade-off between efficiency and accuracy, and reveals an emergent phenomenon where "thinking" and "non-thinking" modes naturally arise. Our current approach relies on output entropy derived from model-generated samples, which may be sensitive to the quality and diversity of candidate outputs; future work will explore more robust uncertainty estimation techniques and broader clinical validation.

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

# A   APPENDIX

## A.1   ETHICS STATEMENT

The present study does not involve any human participants, animal subjects, or the use of personally identifiable information. All datasets employed are obtained from open-access repositories. The data usage, methodological design, and reported findings do not present any anticipated ethical risks. All work has been carried out in line with the ICLR Code of Ethics and adheres strictly to the relevant research integrity guidelines.

## A.2   REPRODUCIBILITY STATEMENT

We place strong emphasis on reproducibility in this work. Detailed descriptions of experimental configurations—including model architectures, hyperparameter settings, and training strategies—are provided in Appendix B.2, with further implementation specifics available in Appendix B.3. To further support transparent validation, we release an anonymous repository containing the complete source code and step-by-step instructions for replicating all experiments presented in this paper.

## A.3   DECLARATION OF LLM USE

Large language models (LLMs) were employed solely for language refinement. They assisted in polishing grammar and clarity of the manuscript text, without contributing to the research design, data analysis, or interpretation.

# B   EXPERIMENTAL SETUP

## B.1   TRAINING RL ALGORITHM

**Group Relative Policy Optimization (GRPO).** To optimize the efficient reasoning objective in Eq. equation 1, both critic-based reinforcement learning methods (e.g., PPO) and critic-free methods (e.g., GRPO (Guo et al., 2024)) can be applied. Considering its simplicity and effectiveness, GRPO has been widely adopted in recent works aiming to enhance the reasoning ability of large language models. Given an input prompt $\mathbf{x}$, a group of $G$ candidate outputs $\{o_i\}_{i=1}^{G}$ is sampled from the current policy $\pi_\theta$, where each $o_i = \{o_{i,1}, \ldots, o_{i,|o_i|}\}$ corresponds to one possible output sequence $\mathbf{y}$. We reformulate the objective of maximizing the expected reward $R(\mathbf{y}, \mathbf{y}^*)$ in Eq. equation 1 as minimizing the following GRPO loss:

$$\mathcal{L}_{\text{GRPO}}(\theta) = -\frac{1}{\sum_{i=1}^{G}|o_i|}\sum_{i=1}^{G}\sum_{t=1}^{|o_i|}\left[\min\left(\frac{\pi_\theta(o_{i,t}\mid\mathbf{x},o_{i,<t})}{\pi_{\theta_{\text{old}}}(o_{i,t}\mid\mathbf{x},o_{i,<t})}\hat{A}_{i,t},\right.\right.$$
$$\left.\left.\text{clip}\left(\frac{\pi_\theta(o_{i,t}\mid\mathbf{x},o_{i,<t})}{\pi_{\theta_{\text{old}}}(o_{i,t}\mid\mathbf{x},o_{i,<t})}, 1-\epsilon,\, 1+\epsilon\right)\hat{A}_{i,t}\right),\right. \tag{8}$$

where $\hat{A}_{i,t}$ denotes the relative advantage of token $o_{i,t}$ within the group, $\pi_{\theta_{\text{old}}}$ represents the policy before the update.

## B.2   BASELINE DETAILS.

We compare AdaThink-Med with strong large language models from both general-purpose and medical domains to enable a comprehensive evaluation. The general-purpose baselines are Qwen-2.5-Instruct-7B and Llama-3.1-Instruct-8B. The medical-domain baselines include MedLlama3, MMed, Med42 (Christophe et al., 2024), OpenBioLLM (Pal & Sankarasubbu, 2024), UltraMedical (Zhang et al., 2024), HuatuoGPT-o1 (Chen et al., 2024b), and m1 (Huang et al., 2025). Length calibration RL based baselines include representative Kimi1.5 (Team et al., 2025), ShortBetter (Yi et al., 2025),DAST (Shen et al., 2025)and CosFn (Yeo et al., 2025). To counteract the length reward hack issue observed in models like Kimi1.5 and ShortBetter, we select the checkpoint from the stable performance period preceding the point where accuracy declines by more than 10% from its peak.

## B.3 IMPLEMENTATION DETAILS.

**Basic Setups.** Training is conducted using 8×H100 GPUs (80GB VRAM each) with PyTorch, leveraging FlashAttention-2 for computational efficiency. GRPO-related experiments are implemented using the `verl` (Sheng et al., 2024) framework to accelerate the training. We adopt the Qwen2.5 (Team, 2024)-7B-Instruct model and Llama3.1-Instruct-8B as the backbone models. The training uses a total batch size of 256, with a constant learning rate of $1e^{-6}$. The KL penalty coefficient is set to $\beta = 0.01$. For inference during GRPO, we deploy the model using `vllm` (Kwon et al., 2023) on 2 GPUs, generating 8 completions per sample, corresponding to the group size $G$ in GRPO. The temperature for `vllm` sampling is set to 1.0. As for our staged adaptive training, we first train the model using GRPO for 300 steps until full convergence is achieved. Subsequently, the proposed length calibration method is applied and train for 200 steps.

**Repetition Penalty with N-gram on LLama- Models**. We also observe that the model exhibits length reward hacking, wherein it increases the length of reasoning chains on incorrectly answered hard questions through repetition rather than substantive reasoning. This finding aligns with the observations in Yeo et al. (2025). However, such length reward hacking is not observed in Qwen backbones, a discrepancy that warrants further investigation. In our experiments with LLaMA backbones, we apply the same n-gram repetition penalty as described in their work.

## B.4 DATASET DETAILS

We conduct experiments on six publicly accessible medical and biomedical benchmarks, covering different knowledge domains and levels of reasoning difficulty. The datasets are summarized as follows:

**MedQA (Jin et al., 2021).** This benchmark is derived from the United States medical licensing examination and is designed to test comprehensive clinical knowledge across a wide range of medical specialties. The standard test split is used for evaluation.

**MedMCQA (Pal et al., 2022).** Originating from Indian medical entrance examinations, this dataset consists of multiple-choice questions that focus on fundamental medical concepts. Evaluation is performed on the official test set provided by the dataset.

**PubMedQA (Jin et al., 2019).** A biomedical question answering dataset constructed from PubMed abstracts. Each sample requires choosing between "yes," "no," or "maybe" to reflect factual understanding in biomedical research literature. We adopt the official test partition for all experiments.

**MMLU-ProM (Wang et al., 2024).** This is the medical subset of the general MMLU benchmark, specifically targeting professional-level medical and health-related knowledge. Following prior work (Chen et al., 2024b), we employ the standard split configuration defined for this subset.

**GPQA-M (Rein et al., 2024).** This dataset represents the biomedical branch of the Graduate-Level Question Answering benchmark. Its questions are manually curated to avoid superficial retrieval and to encourage deeper reasoning. We report results using the official evaluation split.

**MedXpert (Zuo et al., 2025).** A challenging benchmark built to assess advanced medical knowledge, complex reasoning, and clinical decision-making skills. It contains board-style exam questions across multiple body systems and specialties, curated by domain experts to ensure reliability and difficulty.

## B.5 ADDITIONAL EXPERIMENTAL RESULTS

Table 4: Experimental results on using adaptive thinking methods on non-R1 reasoning models, here we use HuatuoGPT-o1, which first SFT with CoT data and then RL with PPO.

| Model | MedQA Acc. | MedQA Len. | MedMCQA Acc. | MedMCQA Len. | PubMedQA Acc. | PubMedQA Len. | MMLU-ProM Acc. | MMLU-ProM Len. | GPQA-M Acc. | GPQA-M Len. | MedXpert Acc. | MedXpert Len. | Avg Acc. | Avg Len. | AES |
|---|---|---|---|---|---|---|---|---|---|---|---|---|---|---|---|
| HuatuoGPT-o1-8B | 76.35 | 568 | 62.82 | 445 | 79.80 | 445 | 63.71 | 521 | 54.35 | 585 | 17.06 | 601 | 59.02 | 527 | – |
| Kimi1.5 (Team et al., 2025) | 71.01 | 373 | 62.87 | 302 | 77.40 | 288 | 67.04 | 340 | 50.26 | 401 | 14.49 | 363 | 57.18 | 345 | 0.19 |
| CosFn (Yeo et al., 2025) | 68.42 | 260 | 62.63 | 126 | 77.30 | 159 | 65.60 | 220 | 48.97 | 187 | 14.24 | 246 | 56.20 | 200 | 0.38 |
| DAST (Shen et al., 2025) | 71.17 | 730 | 61.01 | 619 | 76.70 | 578 | 64.36 | 697 | 45.90 | 794 | 15.02 | 765 | 55.69 | 697 | -0.60 |
| ShortBetter (Yi et al., 2025) | 71.64 | 336 | 63.18 | 271 | 78.40 | 258 | 66.12 | 299 | 49.74 | 353 | 14.45 | 321 | 57.26 | 306 | 0.27 |
| **AdaThink-Med-HuatuoGPT-o1** | 72.74 | 183 | 63.54 | 192 | 76.60 | 72 | 66.96 | 204 | 50.77 | 344 | 13.88 | 221 | 57.41 | 203 | 0.48 |

---

**Algorithm 1** AdaThink-Med: Uncertainty-Guided adaptive thinking

---

**Require:** Dataset $\mathcal{D}$, initial policy $\pi_\theta$, reference policy $\pi_{\text{ref}}$, hyperparameters $\lambda, \beta, \tau, K, \alpha$
**Ensure:** Optimized adaptive policy $\pi_\theta$
 1: **Stage 1: Standard reasoning training (without length calibration)**
 2: **for** epoch = 1 to $N_1$ **do**
 3:      Sample batch $\mathcal{B} \sim \mathcal{D}$
 4:      Compute standard GRPO loss (Eq. 8) with $\mathcal{R}_i = \mathcal{R}_{\text{acc}}^i + \mathcal{R}_{\text{format}}^i$
 5:      Update $\pi_\theta$ via gradient descent
 6: **end for**
 7: **Stage 2: adaptive thinking with length calibration**
 8: Initialize EMA statistics: $\theta_{\mathcal{B}}, \bar{L}_s, \bar{L}_h$
 9: **for** epoch = 1 to $N_2$ **do**
10:      Sample batch $\mathcal{B} \sim \mathcal{D}$
11:      **for** each question $q \in \mathcal{B}$ **do**
12:          Generate $G$ responses $\{o_i\}_{i=1}^G \sim \pi_\theta(\cdot|\mathbf{x}_q)$
13:          Compute token entropies $H_t$ via Eq. 3
14:          Compute trajectory uncertainties $\mathcal{H}_i$ (top-$K$ mean)
15:          Normalize uncertainties $\widetilde{\mathcal{H}}_i$ within batch
16:          Estimate difficulty $\mathcal{D}_q$ via Eq. 5
17:      **end for**
18:      Update $\theta_{\mathcal{B}} \leftarrow \alpha \cdot \text{Quantile}(\{\mathcal{D}_q\}, \tau) + (1 - \alpha) \cdot \theta_{\mathcal{B}}$
19:      Update $\bar{L}_s, \bar{L}_h$ via EMA of simple/hard question lengths
20:      **for** each response $o_i$ in batch **do**
21:          Compute length reward $\mathcal{R}_{\text{len}}^i$ based on $\mathcal{D}_q, \theta_{\mathcal{B}}$
22:          $\mathcal{R}_i \leftarrow \mathcal{R}_{\text{acc}}^i + 0.5\mathcal{R}_{\text{format}}^i + 0.5\mathcal{R}_{\text{len}}^i$
23:      **end for**
24:      Compute GRPO loss $\mathcal{L}_{\text{GRPO}}$ with rewards $\{\mathcal{R}_i\}$
25:      Update $\pi_\theta$ to minimize $\mathcal{L}_{\text{GRPO}} + \beta D_{\text{KL}}(\pi_\theta \| \pi_{\text{ref}})$
26: **end for**

---

**Effectiveness of AdaThink-Med on non-R1-like models.** The adaptive thinking experiments in the main paper are performed on R1-like models, where reasoning capability is acquired through RL training with GRPO. To further validate our approach, we conduct additional experiments on other reasoning models. Our implementation follows the official setup of HuatuoGPT-o1 (Chen et al., 2024b), which employs a large language model (LLM) as a judgment mechanism. Specifically, we use HuatuoGPT-o1-7B (Chen et al., 2024b) as our base model, which is built upon the Qwen2.5-Instruct-7B backbone and trained in two stages: first through supervised fine-tuning (SFT) on 20,000 long-form chain-of-thought (CoT) samples, followed by reinforcement learning via proximal policy optimization (PPO) on 20,000 verifiable medical question-answering samples. All 40,000 samples are converted into open-ended QA format and sourced from the training splits of MedQA-USMLE and MedMCQA. The 20,000 PPO samples are further utilized in our subsequent adaptive GRPO training. For verification, we adopt the distilled verifier released by HuatuoGPT-o1, which is trained using 20,000 scoring samples distilled from GPT-4o.

We compare several RL-based length calibration methods, and the results are summarized in Table 4. Our findings demonstrate that AdaThink-Med achieves the highest efficiency improvement. Specifically, AdaThink-Med reduces the average output length by a factor of 2.6, from 527 to 203, while incurring only an average performance degradation of 1.59%. In comparison, the second-best method, CosFn, compresses the average output length from 527 to 200, but with a larger performance degradation of 2.78%. Overall, these results confirm the effectiveness of AdaThink-Med in enhancing the inference efficiency of diverse reasoning models.

**Estimated Difficulty vs. Response Length Analysis.** Additionally, we have included the requested plot of response length vs. estimated difficulty in Fig. 6. The overall trend observed is as follows: for correct test samples, as the response length increases, the uncertainty also rises, indicating an increase in difficulty. For incorrect samples, both short and long response lengths show high difficulty, with short responses exhibiting a more pronounced rise in uncertainty. This suggests that more reasoning or cognitive effort is required in such cases. Notably, in the region of medium re-

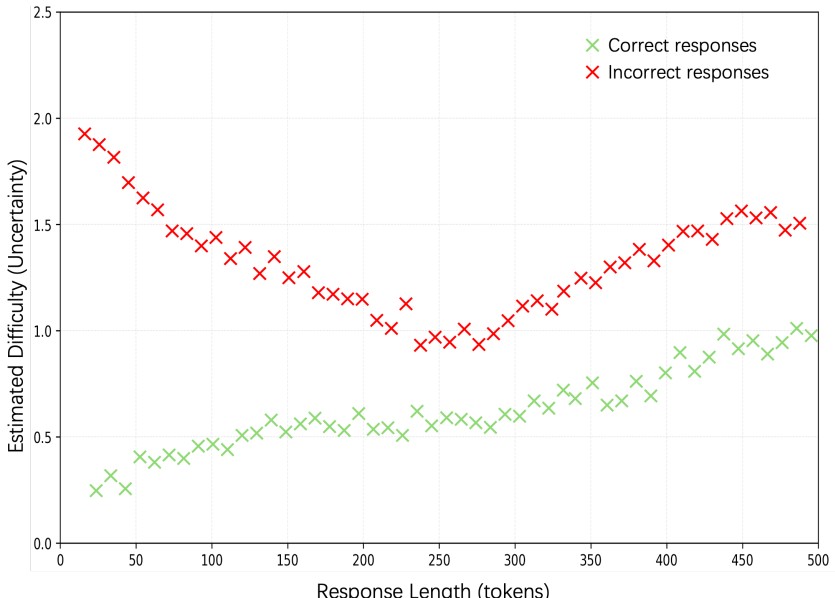

Figure 6: Estimated difficulty vs. response length.

sponse lengths, both uncertainty measures level off, indicating that these problems are challenging to optimize using a single mode.

Table 5: Human evaluation results.

| Method | Response Accuracy (%) | Redundancy Tolerance (%) |
|---|---|---|
| Kimi | 71.25 | 73.00 |
| CosFn | 70.34 | 67.00 |
| DAST | 68.37 | 65.00 |
| ShortBetter | 70.23 | 72.00 |
| AdaThink-Med | **76.25** | **89.00** |

**Human Evaluation.** The core assumption behind our method is that models need to engage in sufficient reasoning before arriving at the correct answer. The adaptive algorithm ensures that the maximum reduction in reasoning length corresponds to the model's required minimum reasoning threshold. However, the authenticity of thinking traces should be validated if missing some important steps due to adaptive training, which is crucial for clinical real-world application. To address this, we invited three clinicians to compare the algorithm's output on 500 randomly sampled test cases, focusing on both the accuracy of the responses and the acceptability of reasoning redundancy (i.e., whether the reasoning included a minimum amount of reliable and correct reasoning content). As shown in Table 5, our AdaThink-Med received significantly more votes for superior generation quality compared to other methods, demonstrating both correctness and fidelity in its reasoning traces, as perceived by human evaluators, confirming the model's improved performance and reasoning authenticity.

**Robustness to Low-Quality Outputs**: To assess the robustness of our algorithm to low-quality outputs, we conducted experiments using a low-quality dataset. Specifically, after training the Qwen2.5-Instruct-7B model with GRPO, we performed multiple rounds of response reasoning for the same question in the training set. We then ranked the samples based on accuracy and selected both the fully incorrect samples and some low-accuracy samples to form our low-quality dataset. These samples, which generate high uncertainty, present a significant challenge for training. As shown in Table 6, the experimental results indicate that AdaThink-Med consistently outperforms other models under these challenging conditions, demonstrating the robustness of our algorithm.

Table 6: Experimental results showing the robustness of AdaThink-Med under low-quality output conditions.

| Method | PubMedQA | | MMLU-ProM | | GPQA-M | | MedXpert | | Avg. | |
|---|---|---|---|---|---|---|---|---|---|---|
| | Acc. | Len. | Acc. | Len. | Acc. | Len. | Acc. | Len. | Acc. | Len. |
| Qwen-GRPO | 72.30 | 396 | 66.12 | 535 | 48.46 | 575 | 14.48 | 592 | 50.34 | 524 |
| Kimi | 71.20 | 230 | 65.23 | 328 | 46.52 | 376 | 14.31 | 366 | 49.34 | 325 |
| CosFn | 70.90 | 301 | 64.17 | 428 | 47.66 | 351 | 13.87 | 353 | 49.15 | 358 |
| DAST | 72.30 | 298 | 64.88 | 535 | 47.35 | 507 | 13.26 | 498 | 49.45 | 460 |
| ShortBetter | 70.50 | 354 | 63.15 | 297 | 45.23 | 330 | 13.26 | 320 | 48.04 | 325 |
| AdaThink-Med | 72.10 | 89 | 65.38 | 115 | 47.51 | 212 | 14.33 | 108 | 49.83 | 131 |

**VQA Generalization Ability**: To evaluate the effectiveness of AdaThink-Med in multi-modal scenarios, we conducted a clinical free-text generation task. Specifically, we included experimental results from the radiology (X-ray) report generation task, which is a Visual Question Answering (VQA) task. The MIMIC-CXR-2.0 (Johnson et al., 2019) dataset was used for validation. In the test set, we employed LLM-as-judge to assess the generated reports and compare them to the ground truth in two key aspects: the completeness of the report and its conciseness. As shown in Table 7, the results demonstrate that AdaThink-Med effectively reduces the report length while maintaining completeness, highlighting its exceptional performance.

Table 7: Experimental results for the radiology report generation task. Completeness is measured by the extent to which the generated report covers all required aspects, and redundancy reflects the level of unnecessary components in the generated text.

| Method | Completeness ↑ | Redundancy ↓ |
|---|---|---|
| Kimi | 66.2 | 42.5 |
| CosFn | 34.5 | 54.1 |
| DAST | 46.7 | 50.7 |
| ShortBetter | 55.3 | 45.2 |
| AdaThink-Med | **78.4** | **22.5** |

**Sensitivity Analysis of TOP-$K$**: To assess the impact of the hyperparameter TOP-$K$, we conducted ablation experiments using the Qwen2.5-Instruct-7B model. The results shown in Table 8 indicate that variations in the TOP-$K$ value lead to stable efficiency improvements in AdaThink-Med. However, both very large and very small values of TOP-$K$ result in suboptimal efficiency gains. This occurs because smaller $K$ values reduce the number of sampled tokens, leading to inaccurate difficulty estimations, while larger $K$ values introduce an averaging effect that dilutes the model's entropy due to the presence of plain tokens.

Table 8: Ablation results on the TOP-$K$ hyperparameter for entropy calculation.

| $K$ | MedQA | | MedMCQA | | PubMedQA | | MMLU-ProM | | GPQA-M | | MedXpert | | Avg. | | AES |
|---|---|---|---|---|---|---|---|---|---|---|---|---|---|---|---|
| | Acc. | Len. | Acc. | Len. | Acc. | Len. | Acc. | Len. | Acc. | Len. | Acc. | Len. | Acc. | Len. | |
| - | 63.86 | 490 | 65.68 | 359 | 73.10 | 386 | 62.23 | 523 | 46.41 | 562 | 12.28 | 585 | 52.44 | 484 | - |
| 5 | 67.83 | 102 | 67.53 | 125 | 74.22 | 87 | 65.93 | 131 | 46.96 | 221 | 13.95 | 102 | 54.41 | 128 | +0.85 |
| 20 | 68.34 | 79 | 68.74 | 109 | 73.50 | 62 | 66.40 | 121 | 48.72 | 175 | 14.16 | 109 | 54.99 | 106 | +0.93 |
| 50 | 67.55 | 141 | 68.35 | 127 | 73.92 | 105 | 68.32 | 143 | 46.79 | 190 | 14.24 | 117 | 54.86 | 137 | +0.86 |

## B.6 QUALITATIVE RESULTS

Fig. 7 presents a straightforward evaluation of basic genetics knowledge. The baseline models, m1 and HuatuoGPT-o1, display evident overthinking through numerous redundant reasoning steps, consuming 766 and 451 tokens respectively. In contrast, AdaThink-Med achieves a correct solution with only 45 tokens, demonstrating conciseness and accuracy. Furthermore, the baselines frequently produce incorrect conclusions during analysis and rely on a "wait" mechanism to initiate unnecessary rethinking cycles.

Table 9: Dataset selection experimental results. $\delta$ represents the ratio of training datasets.

| $\delta$ | In-domain | | | | Out-domain | | | | | | | | Avg. | |
|---|---|---|---|---|---|---|---|---|---|---|---|---|---|---|
| | MedQA | | MedMCQA | | PubMedQA | | MMLU-ProM | | GPQA-M | | MedXpert | | Acc. | Len. |
| | Acc. | Len. | Acc. | Len. | Acc. | Len. | Acc. | Len. | Acc. | Len. | Acc. | Len. | | |
| 0% | 63.86 | 490 | 56.68 | 359 | 73.10 | 386 | 62.28 | 523 | 46.41 | 562 | 12.28 | 585 | 52.44 | 484 |
| 20% | 64.81 | 497 | 58.50 | 373 | 74.40 | 400 | 64.82 | 526 | 48.21 | 565 | 14.00 | 585 | 54.12 | 491 |
| 40% | 65.44 | 512 | 58.81 | 367 | 74.30 | 396 | 66.51 | 522 | 51.28 | 564 | 14.37 | 590 | 55.12 | 492 |
| 60% | 68.50 | 488 | 58.69 | 356 | 73.10 | 381 | 66.12 | 513 | 47.44 | 535 | 13.96 | 569 | 54.63 | 474 |
| 80% | 68.42 | 504 | 58.38 | 370 | 72.30 | 388 | 67.49 | 527 | 50.00 | 576 | 14.73 | 579 | 55.22 | 491 |
| 100% | 67.55 | 504 | 59.52 | 378 | 72.30 | 396 | 66.12 | 535 | 48.46 | 575 | 14.48 | 592 | 54.74 | 497 |

Figure 7: Case 1. AdaThink-Med delivers correct answer while consumes far fewer tokens than m1-7B and HuatuoGPT-o1-8B.

A 51-year-old man presents the emergency room with chest pain. He mentions that the pain started several hours ago and radiates to his left neck and shoulder. He also mentions that he has some difficulty in breathing. He says that he has had similar chest pains before, but nothing seemed to be wrong at that time. He was diagnosed with high cholesterol during that episode and was prescribed medication. He also has a 3-year history of gastritis. The blood pressure is 130/80 mm Hg, respirations are 18/min, and the pulse is 110/min. He seems a little anxious. The physical examination reveals no significant abnormalities. An ECG shows slight changes in the leads. His physician talks to him about the benefits of taking low-dose aspirin daily. Which of the following would be a contraindication to the use of aspirin in this patient? Options: A: ECG changes B: Hypercholesterolemia C: Gastritis D: Increased pulse rate.

#346 tokens (UltraMedical-8B-3.1)

To address the question about contraindications for the use of low-dose aspirin in this patient, we need to consider each option in the context of known risks associated with aspirin therapy:\n\nA. **ECG changes**: Aspirin is commonly used in the setting of ischemic heart disease due to its antiplatelet effects, which can prevent clot formation. ECG changes could suggest underlying heart disease, such as angina or myocardial infarction, where aspirin might be beneficial rather than contraindicated.\n\nB. **Hypercholesterolemia**: High cholesterol itself is not a direct contraindication to aspirin therapy. However, the use of aspirin in patients with hypercholesterolemia is more related to cardiovascular risk reduction rather than a direct contraindication.\n\nC. **Gastritis**: Aspirin can exacerbate gastrointestinal issues because it inhibits prostaglandin synthesis, which protects the gastric lining. In patients with a history of gastritis, aspirin can increase the risk of gastrointestinal bleeding. This makes aspirin potentially harmful in the context of existing gastrointestinal conditions.\n\nD. **Increased pulse rate**: While aspirin can occasionally cause an increase in heart rate, this is not a contraindication for its use. The primary concern with increased pulse rate would be the potential for underlying cardiac issues, but this does not directly contraindicate aspirin use.\n\nGiven these considerations, the most relevant contraindication among the options provided is the patient's history of gastritis (C), as aspirin can exacerbate gastrointestinal bleeding, especially in someone who already has a compromised gastric lining.\n\n**So, the answer is C.** √

#496 tokens (HuatuoGPT-o1-8B)

## Thinking\n\nAlright, let's think about this. We've got a 51-year-old guy who's come into the ER with chest pain. He says it started a few hours ago and it's spreading to his left neck and shoulder. That sounds like classic heart issues to me. And he's having trouble breathing too. Not good.\n\nHe's had these chest pains before, but nothing showed up back then. They just told him he had high cholesterol and gave him meds for it. Okay, so maybe it's something new or getting worse. And he's been dealing with gastritis for three years. That's not great, but let's see how it plays into things.\n\nHis vitals are interesting. Blood pressure is okay at 130/80, but his heart rate is up at 110. That could mean he's anxious or there's something more going on. The doctor mentioned something about slight ECG changes. Hmm, that makes me think we're definitely looking at some kind of cardiac issue here.\n\nNow, the doctor wants to put him on low-dose aspirin. That's a common move when you're worried about a heart attack. But wait, what about the gastritis? Aspirin can really irritate the stomach lining, especially if someone already has gastritis. That's a big concern because it could make his condition worse.\n\nLet's think about the other stuff. High cholesterol is a risk factor for heart disease, so aspirin might actually help with that. And an increased pulse rate could be a sign of stress or anxiety, but it's not a dealbreaker for aspirin. So, those aren't reasons to say no to aspirin.\n\nThe ECG changes are a bit tricky. They suggest something's up with the heart, which is why they want to start aspirin. It's not a reason to avoid it; it's a reason to give it.\n\nSo, putting it all together, the big red flag here is the gastritis. Aspirin can cause stomach problems, and with his history of gastritis, it's not a good idea. Yeah, that's the one that stands out as a contraindication.\n\n## Final Response\n\nThe contraindication to the use of aspirin in this patient is C: Gastritis. Aspirin can exacerbate gastric issues, particularly in someone with a history of gastritis, making it a significant concern in this case. √

#49 tokens (AdaThink-Qwen)

The most common contraindication for aspirin use in patients with a history of gastrointestinal issues like gastritis is the risk of exacerbating these conditions due to aspirin's potential to cause gastric irritation or bleeding.\n\n\\boxed{C} √

Figure 8: Case 2. AdaThink-Med delivers correct answer while consumes far fewer tokens than UltraMedical-8B-3.1 and HuatuoGPT-o1-8B.

A 31-year-old man comes to the emergency department because of acute onset neck pain and enlargement. Specifically, he reports that he has been experiencing pain and swelling of the anterior portion of his neck near the midline. Otherwise, he says that he has been getting tired easily and feels cold often. Physical exam reveals a painful diffusely enlarged thyroid gland with many small nodules. A biopsy is obtained showing diffuse cellular hyperplasia with areas of focal colloid hyperplasia. Given these findings, the patient is started on appropriate therapy, and the neck mass becomes smaller over time. Which of the following is most likely associated with the cause of this patient's symptoms?\nOptions:\nA: HLA-B8 risk factor\nB: Iodine deficiency\nC: Presence of embryological remnant\nD: Proliferation of fibroinflammatory cells

#### #406 tokens (UltraMedical-8B-3.1)

The clinical presentation of acute onset neck pain and enlargement of the thyroid gland, along with symptoms of fatigue and feeling cold, suggests a thyroid dysfunction. The physical examination finding of a diffusely enlarged thyroid gland with many small nodules points towards a thyroiditis or a goiter rather than a solitary nodule.\n\nThe biopsy result showing diffuse cellular hyperplasia with areas of focal colloid hyperplasia further supports the diagnosis of a thyroiditis, specifically a form that involves the entire gland (diffuse). This can be consistent with Hashimoto's thyroiditis, which is an autoimmune condition and can present with these findings. However, the acute onset and painful nature of the enlargement are more suggestive of subacute thyroiditis (also known as de Quervain's thyroiditis), which is often associated with a viral infection.\n\nSubacute thyroiditis is not typically associated with HLA-B8 (which is more commonly associated with autoimmune conditions like type 1 diabetes and celiac disease) or the presence of an embryological remnant (which would suggest a congenital or developmental anomaly leading to a thyroglossal duct cyst or ectopic thyroid tissue).\n\nIodine deficiency is a common cause of goiter, but it typically does not cause a painful thyroid gland and would not explain the acute presentation. It also tends to cause a symmetrical enlargement of the thyroid rather than multiple nodules.\n\nProliferation of fibroinflammatory cells is characteristic of subacute thyroiditis, where there is a granulomatous inflammation due to the infiltration of the thyroid gland by inflammatory cells, including lymphocytes, plasma cells, and macrophages.\n\nGiven the clinical presentation and biopsy findings, the most likely cause of the patient's symptoms is subacute thyroiditis. The patient's symptoms and the response to treatment (with the neck mass becoming smaller over time) support this diagnosis. Subacute thyroiditis often resolves on its own, although symptomatic treatment may be necessary during the acute phase. \n\nSo, the answer is D. ✗

#### #436 tokens (HuatuoGPT-o1-8B)

## Thinking\n\nAlright, so we have a 31-year-old guy who's suddenly got neck pain and swelling right in the front, near the middle. He's also feeling really tired and cold a lot. That's interesting. When we check him out, his thyroid is all swollen and lumpy, like there are lots of little nodules. They did a biopsy, and it shows a bunch of cells growing too much and some areas with extra colloid. This sounds like a classic case of Graves' disease. \n\nGraves' disease is known for causing the thyroid to get all overactive and swell up. It's an autoimmune thing where the body's immune system goes a bit haywire and starts attacking the thyroid. So, it makes sense that he'd feel tired and cold because his metabolism is probably going into overdrive. \n\nNow, let's think about what might be causing this. Graves' disease is often linked to certain genetic factors. People with HLA-B8 are more prone to developing it. That's a specific genetic marker that increases the risk. \n\nBut wait, there's also iodine deficiency to consider. It can affect the thyroid, but it usually causes a different kind of problem, more like a goiter or hypothyroidism, not hyperthyroidism. So, that doesn't seem to fit here. \n\nWhat about embryological remnants? Those are more about things like thyroglossal duct cysts, which don't really match our guy's situation. And fibroinflammatory cells? That's more about inflammation, not the kind of cellular hyperplasia we're seeing here. \n\nSo, putting it all together, the most likely cause of his symptoms is the HLA-B8 risk factor. It's a genetic predisposition that fits with the autoimmune nature of Graves' disease. Yeah, that makes sense.\n\n## Final Response\n\nThe most likely association with the cause of this patient's symptoms is **A: HLA-B8 risk factor**. The symptoms and biopsy findings suggest Graves' disease, an autoimmune condition that often has a genetic component, with HLA-B8 being a known risk factor. ✗

#### #156 tokens (AdaThink-Qwen)

Based on the clinical presentation, physical exam, and biopsy findings, the most likely diagnosis is Hashimoto's thyroiditis, which can present with a painful, diffusely enlarged thyroid with nodularity and colloid hyperplasia.\n\nThe correct association is:\n\nB: Iodine deficiency\n\nHowever, iodine deficiency is more commonly associated with goiter formation without the specific symptoms described, so it is less likely given the specific symptoms presented. The most probable cause is an autoimmune disorder leading to Hashimoto's thyroiditis.\n\nGiven options and the context, the closest association for the cause would be related to immune-mediated disease.\n\nFinal answer: \\boxed{B} (though iodine deficiency directly isn't the most common direct trigger, B is the closest among the options provided). ✓

Figure 9: Case 3. An overthinking case of UltraMedical-8B-3.1 and HuatuoGPT-o1-8B.

Intestinal lactase deficiency leads to lactose maldigestion which subsequently can lead to lactose intolerance, i.e. causing clinical symptoms. Infants and children normally express sufficient lactase levels as not to experience clinical symptoms. Overall, adults can be classified as lactase-persisters or lactase non-persisters. What is the approximate percentage of Lactase non-persisters worldwide?\n\nOptions:\nA: 30%-35%\nB: 80%-85%\nC: 90%-95%\nD: 55%-60%\nE: 5%-10%\nF: 45%-50%\nG: 60%-65%\nH: 10%-15%\nI: 20%-25%\nJ: 70%-75 %\n

**#1926 tokens (m1-7B)**

think\nOkay, let's see. The question is asking about the approximate percentage of lactase non-persisters worldwide. Hmm. I remember that lactase non-persister is the condition where people lose the ability to produce lactase after infancy, leading to lactose intolerance. So, lactase persisters are those who can digest lactose into adulthood.\n\nFrom what I've studied before, this varies a lot depending on ethnicity and geography. For instance, in populations with a history of dairying, like Northern European descent, lactase persistence is more common. Conversely, in many Asian, African, and South American populations, lactase non-persistence is more prevalent because there wasn't as much selective pressure to retain that ability.\n\nThe options given are percentages between 5% to 95%, so it's about which broad range is correct. Let me recall some numbers. I think that around 60-70% of the world's population becomes lactase non-persistent after infancy. That would put it in the range of option J: 60-65%, but wait, option J says 70-75%. Wait, maybe I should check again.\n\nWait, another angle: in many studies, it's often cited that about 70% of the global population has lactase non-persistance. For example, in regions like East Asia, it's almost everyone, while in Europe, it's around 20%. Merged globally, I think the average is roughly 70% or higher. But sometimes different sources might say slightly different numbers.\n\n······way. The World Health Organization (WHO) might have data on this, but I don't have access right now. From my prior education, I remember that the majority of adults globally are lactase non-persisters, with estimates around 70%. Therefore, I think option J:70-75% is correct. But to confirm, perhaps cross-checking with some percentages.\n\nFor example, in a study from the National Academy of Sciences, it states that approximately 75% of the world's population lacks the ability to fully digest lactose after infancy. So that would support option J. However, if the exact number is 70%, then J includes that. Therefore, the answer is J. However, if another source says 65%, then G. But without precise data, relying on commonly cited figures. Since many sources point to around 70-75%, I will go with J.\nanswer\nAnswer: The percentage of lactase non-persisters worldwide refers to individuals who lose lactase production after infancy, leading to lactose intolerance. This condition varies significantly by population, with higher rates in regions lacking a historical context of dairy consumption (e.g., Asia, Africa) and lower rates in lactase-persister populations (e.g., Northern Europe). \n\nGlobal estimates average around **70–75%** of adults being lactase non-persisters, supported by widespread references in human biology and medical literature. While sub-regional variations exist (e.g., >90% in East Asia, ~20% in Northern Europe), the overall global average aligns closely with **70–75%**, making **Option J** the correct choice.\n\n\\boxed{J} ✗

**#453 tokens (HuatuoGPT-o1-8B)**

## Thinking\n\nOkay, let's think about this. So, we're looking at lactase deficiency and how it relates to lactose intolerance. Lactase is that enzyme that helps break down lactose in milk, right? If you don't have enough of it, you might feel pretty uncomfortable after eating dairy.\n\nNow, infants and young kids usually have plenty of lactase, so they don't really have issues with milk. But as people grow up, some of them start to lose this enzyme. That's when we get these two groups: lactase-persisters and lactase non-persisters. The persisters keep making lactase into adulthood, while the non-persisters don't.\n\nSo, what's the deal with the numbers? I've heard that a lot of people around the world are lactase non-persisters. It's a pretty common thing, especially in certain parts of the world like Africa, Asia, and parts of Europe. But how many exactly?\n\nHmm, I remember reading that somewhere between 70% and 80% of the world's population falls into this category. That seems like a big chunk of people. But wait, I should double-check that number. It's always good to verify.\n\nLet me think. Some sources say it's even higher, like 90% in some places. But I'm not sure if that's right. I need to dig deeper.\n\nOh, here's something interesting. In some regions like Europe and North America, it's much lower, like around 5-10%. That's because those populations have more dairy in their diets, so there's more selective pressure to keep the lactase gene active.\n\nSo, if I put all this together, it seems like globally, the percentage of lactase non-persisters is somewhere between 70% and 85%. Yeah, that feels right. It's a pretty high number, but it makes sense given the global distribution of lactase activity.\n\nAlright, I'm confident with that answer. The approximate percentage of lactase non-persisters worldwide is likely around 80-85%.\n\n## Final Response\n\nThe approximate percentage of lactase non-persisters worldwide is B: 80%-85%. √

**#116 tokens (AdaThink-Qwen)**

To determine the approximate percentage of lactase non-persisters worldwide, we need to consider the global distribution of this trait. It's well-established that lactase persistence is more common in populations with a history of cattle domestication, such as Northern Europe, while lactase non-persistence is more common in other regions.\n\nHowever, for a general estimate without specific regional focus, the most widely cited global prevalence of lactase non-persistence is around 70-80%. This aligns best with option J (70%-75%).\n\n\\boxed{J} √

Figure 10: Case 4. An overthinking case of HuatuoGPT-o1-8B.

## C  MORE RELATED WORKS

**Medical LLMs.** With rapid advancements in training data and synthetic data formulation, LLMs in the medical domain have achieved significant performance improvements. Notable developments include instruction-tuned medical chatbot series (Pal & Sankarasubbu, 2024; Qiu et al., 2024; Christophe et al., 2024; Zhang et al., 2024), reasoning models incorporating Chain-of-Thought (CoT) methodologies (Chen et al., 2024b; Huang et al., 2025; Rui et al., 2025b), and agent collaboration (Chen et al., 2025a). These models have seen substantial adoption in both academic research and real-world applications. However, for clinical deployment, computational efficiency remains a critical concern alongside diagnostic accuracy. While SynapseRoute (Zhang et al., 2025) explores a router network to partition LLMs into "thinking" and "non-thinking" pathways, this approach requires additional manual labeling to train difficulty-classification networks, incurring extra computational and human resource costs. In contrast, we propose an efficient method that autonomously allocates computational resources without requiring supplementary modules or human intervention, thereby expanding potential application domains.

