# OpenReview forum: "AdaThink-Med: Medical Adaptive Thinking with Uncertainty-Guided Length Calibration"
_ICLR.cc/2026/Conference — Submitted to ICLR 2026_

### Official Review · Reviewer_52Pp · 2025-10-27

**Soundness:** 3
**Presentation:** 3
**Contribution:** 3
**Rating:** 6
**Confidence:** 3

**Summary:**

This paper proposes AdaThink-Med, the first end-to-end adaptive thinking framework for medical large language models (LLMs), addressing the critical trade-off between reasoning performance and inference cost in clinical applications. Complemented by momentum-based smoothing and staged training, AdaThink-Med achieves up to 6.4× average length reduction across six public medical QA benchmarks with only minimal performance degradation. Key contributions include: (1) the first end-to-end adaptive framework for medical LLMs with autonomous "thinking/non-thinking" mode switching; (2) entropy-based uncertainty quantification for difficulty estimation; (3) a high-quality dataset selection strategy leveraging length distribution; (4) state-of-the-art efficiency-performance balance on medical benchmarks.

**Strengths:**

1. AdaThink-Med is the good end-to-end adaptive thinking framework tailored for medical LLMs, eliminating the need for manual difficulty labeling and enabling dynamic mode switching without separate classifiers.
2. The experimental design is rigorous and comprehensive.
3. The work addresses a critical bottleneck for medical LLM deployment: balancing inference efficiency (cost, latency) and diagnostic accuracy.

**Weaknesses:**

1. The framework relies on entropy derived from model-generated outputs, but the paper does not fully validate its robustness to low-quality or biased candidate outputs.
2. The paper treats "problem difficulty" as a general construct but does not explore how medical domain characteristics (e.g., clinical ambiguity, multi-step differential diagnosis, rare diseases) affect difficulty estimation.
3. While the paper compares against many baselines, it lacks direct comparison with state-of-the-art medical LLMs optimized for efficiency.
4. Modern medical AI often requires integrating text (EHRs) and imaging (X-rays, MRIs), but AdaThink-Med is limited to text-based QA.
5. While the performance compensation mechanism addresses greedy length reduction, the paper does not explain how it handles edge cases in medical QA

**Questions:**

1. The top-K token selection in entropy calculation (Section 3.1.1) uses an unspecified K (e.g., top 20%). What is the rationale for choosing K, and have you conducted a sensitivity analysis on K across different medical benchmarks.
2. In staged training, how is the convergence of the first stage (standard GRPO training without length calibration) defined? Does the choice of convergence criterion (e.g., validation accuracy plateau, step count) impact the performance of the second-stage adaptive calibration?
3. The paper observes emergent "thinking/non-thinking" modes—have you analyzed whether these modes correlate with specific medical task types (e.g., basic medical fact QA vs. complex prognosis prediction)?
4. The dataset selection strategy achieves better performance with 40% of the data. What characteristics of the retained 40% data (e.g., difficulty distribution, question type) drive this improvement? Is this finding generalizable to other medical datasets beyond AlphaMed19k?
5. How does AdaThink-Med handle medical questions with high inherent uncertainty (e.g., "What is the most likely diagnosis for a patient with non-specific symptoms") versus questions with low uncertainty but rare answers?
6. Have you evaluated the framework’s performance on subpopulations (e.g., questions about pediatric vs. adult medicine, common vs. rare diseases)? Medical LLMs often exhibit bias in subpopulation performance—does AdaThink-Med’s adaptive strategy mitigate or exacerbate such biases?

---

> ### Author Response · Authors · 2025-11-22
> **Part1**
>
> Thank you very much for taking the time to review and for your support. We try our best to address your questions as follows.
> - **Weakness 1 - robustness to low-quality or biased candidate outputs**: Thank you for your insightful suggestion. We have included experiments with a low-quality dataset. Specifically, after training the Qwen2.5-Instruct-7B model using GRPO, we performed multiple response reasoning for the same question on the training set. **We then ranked the samples by accuracy and selected the fully incorrect samples and some low-accuracy samples as our low-quality dataset, as these samples generate high uncertainty, presenting a great challenge for training**. The experimental results show that AdaThink-Med still maintains significantly superior performance under these extreme conditions, demonstrating the robustness of our algorithm.
>
>     | Model   | PubMedQA Acc. | PubMedQA Len. | MMLU-ProM Acc. | MMLU-ProM Len. | GPQA-M Acc. | GPQA-M Len. | MedXpert Acc. | MedXpert Len. | Avg. Acc. | Avg. Len. |AES|
>     |--------------|---------------|---------------|----------------|----------------|-------------|-------------|----------------|----------------|-----------|-----------|--|
>     | Qwen-GRPO    | 72.30    | 396 | 66.12| 535  | 48.46  | 575    | 14.48| 592  | 50.34| 524  |-|
>     | Kimi    | 71.20    | 230 | 65.23| 328  | 46.52  | 376    | 14.31| 366  | 49.34| 325  |+0.28|
>     | CosFn   | 70.90    | 301 | 64.17| 428  | 47.66  | 351    | 13.87| 353  | 49.15| 358  |+0.20|
>     | DAST    | 72.30    | 298 | 64.88| 535  | 47.35  | 507    | 13.26| 498  | 49.45| 460  |+0.03|
>     | ShortBetter  | 70.50    | 354 | 63.15| 297  | 45.23  | 330    | 13.26| 320  | 48.04| 325  |+0.15|
>     | AdaThink-Med | 72.10    | 89  | 65.38| 115  | 47.51  | 212    | 14.33| 108  | 49.83| 131  |**+0.70**|
>
> - **Weakness 2 - Does not explore how medical domain characteristics affect difficulty estimation**: Thank you for your careful observation. While our validation experiments were conducted in the medical LLM domain, our algorithm is general in nature. **The proposed entropy-based difficulty measure for adaptive reasoning is novel both in the generic domain and in medical contexts**. In the **medical field, there is an urgent need for deployment, with higher requirements for latency and inference costs**, necessitating more efficient model services. **For example, in stroke scenarios, decisions regarding thrombolysis or surgery need to be made in a very short time because every second counts**. In such cases, the model must not only provide accurate results but also operate with high speed. **Our approach represents the first step in adaptive reasoning for medical applications**. We have conducted extensive validation using publicly available medical QA datasets, which demonstrates the effectiveness of our algorithm in addressing these challenges.
>
>     We completely agree with your suggestion to incorporate medical domain characteristics such as clinical ambiguity, multi-step differential diagnosis, and rare diseases into the difficulty estimation process. This is an excellent idea, and we will consider it in our future work.
>
> - **Weakness 3 - Lacks direct comparison with state-of-the-art medical LLMs optimized for efficiency**: Thank you for your valuable feedback. In fact, to the best of our knowledge, **AdaThink-Med is the first end-to-end adaptive reasoning framework in the medical domain**. While there is a non-end-to-end work, **SynapseRoute**[1], **it is not open-source and is trained on private data**, making it impossible to replicate. As a result, there are no existing medical adaptive reasoning models available for direct comparison.
>
>     [1] SynapseRoute: An Auto-Route Switching Framework on Dual-State Large Language Model.
>
>
> - **Weakness 4 - limited to text-based QA**: Thank you for your insightful suggestion. As a complement, we have added a clinical free-text generation task. Specifically, **we included experimental results on the radiology(X-ray) report generation task, which is a Visual Question Answering (VQA) task**. We used the **MIMIC-CXR-2.0** dataset for validation. In the test set, we employed **LLM-as-judge** to evaluate the generated reports and compare them to the ground truth in two aspects: the completeness of the generated report and its conciseness. The results demonstrate that AdaThink-Med effectively compresses the report length while ensuring completeness, showcasing its exceptional effectiveness.
>
>     | Method   | Completeness ↑ | Redundancy ↓ |
>     |---------------|----------------|--------------|
>     | Kimi| 66.2 | 42.5    |
>     | CosFn    | 34.5 | 54.1    |
>     | DAST| 46.7 | 50.7    |
>     | ShortBetter   | 55.3 | 45.2    |
>     | AdaThink-Med  | **78.4** | **22.5**    |

---

> ### Author Response · Authors · 2025-11-22
> **Part2**
>
> - **Weakness 5 - Edge Cases in Medical QA**: Thank you for your valuable feedback. We think the "edge cases" you mentioned in medical QA may refer to two key situations: **whether unnecessarily verbose outputs are generated for hard questions** and **whether important reasoning steps are missed in the process of reducing token usage**. We will address these concerns separately.
>
>     1. **Whether unnecessarily verbose outputs are generated for hard questions**: After training the base model with GRPO, we performed multiple response samplings for each question in the validation set and identified the bottom 20% of samples with the lowest average accuracy as "hard questions." After adaptive training, we compared the changes in output length for this subset, as shown in the table below. The results indicate that **AdaThink-Med does not generate unnecessarily verbose outputs for hard questions**, validating the effectiveness of our approach.
>
>         | Model    | MedQA Acc. | MedQA Len. | MedMCQA Acc. | MedMCQA Len. | PubMedQA Acc. | PubMedQA Len. | MMLU-ProM Acc. | MMLU-ProM Len. | GPQA-M Acc. | GPQA-M Len. | MedXpert Acc. | MedXpert Len. | Avg. Acc. | Avg. Len. |
>         |---------------|------------|------------|--------------|--------------|---------------|---------------|----------------|----------------|-------------|-------------|----------------|----------------|-----------|-----------|
>         | Qwen-GRPO| 56.13 | 876   | 52.68   | 683| 63.90    | 125 | 61.83| 674  | 43.27  | 890    | 13.15| 1033 | 48.49| 714  |
>         | AdaThink-Med  | **58.22** | **129**   | **54.25**   | **253**| **65.60**    | **76**  | **63.97**| **236**  | **46.53**  | **323**    | **14.28**| **567**  | **50.48**| **264**  |
>
>     2. **Whether important reasoning steps are missed in the process of reducing token usage**: Significant token reduction mainly occurs for easier questions. Since key reasoning steps are difficult to quantify, we conducted a **User-centered Evaluation** with clinicians to address this. As shown in the table below, AdaThink-Med demonstrates SOTA redundancy tolerance, indicating that we did not miss important reasoning steps in our efforts to reduce token usage. This can also be explained by our training mechanism: if important reasoning steps were skipped to reduce token usage, the model's entropy would inevitably increase, leading to higher uncertainty. As a result, the difficulty would rise, causing the model to increase output length. Therefore, **we did not reduce token usage at the cost of missing important reasoning steps**.
>
>         | Method   | Response Accuracy (↑)  | Redundancy Tolerance (↑) |
>         |---------------|-------------------|----------------------|
>         | Kimi| 71.25%  | 73%   |
>         | CosFn    | 70.34%  | 67%   |
>         | DAST| 68.37%  | 65%   |
>         | ShortBetter   | 70.23%  | 72%   |
>         | AdaThink-Med  | **76.25%**  | **89%**   |
>
>
> - **Question 1 - Rationale for Choosing $K$ and Sensitivity Analysis**: Thank you for your suggestion. **The value of $K$ used in our paper is consistent with the proportion employed in the large model entropy research paper [1], specifically 20%, as this ratio can represent the overall uncertainty with less noise.** We chose this value because smaller values of $K$ can lead to inaccurate estimations, while larger values are influenced by the entropy of plain text, thereby smoothing out the model's entropy. We have added ablation experiments on the hyperparameter $K$ using the Qwen2.5-Instruct-7B model. The results show that **changes in the $\text{TOP-K}$ value lead to stable efficiency improvements in AdaThink-Med**. However, both very large and very small $\text{TOP-K}$ values result in suboptimal efficiency improvements. The reason for this is that smaller $K$ values lead to fewer sampled tokens, causing inaccurate difficulty estimation, while larger $K$ values cause an averaging effect where the entropy of plain tokens dilutes the model's entropy.
>
>
>     | TOPK| MedQA Acc. | MedQA Len. | MedMCQA Acc. | MedMCQA Len. | PubMedQA Acc. | PubMedQA Len. | MMLU-ProM Acc. | MMLU-ProM Len. | GPQA-M Acc. | GPQA-M Len. | MedXpert Acc. | MedXpert Len. | Avg. Acc. | Avg. Len. | AES   |
>     |--------|------------|------------|--------------|--------------|---------------|---------------|----------------|----------------|-------------|-------------|----------------|----------------|-----------|-----------|-------|
>     | - | 63.86 | 490   | 65.68   | 359| 73.10| 386 | 62.23| 523|46.41| 562|12.28| 585  | 52.44| 484  | -|
>     | 5  | 67.83 | 102   | 67.53| 125| 74.22|87|65.93|131| 46.96|221| 13.95|102|54.41| 128  | +0.85 |
>     | 20 | 68.34 | 79    | 68.74   | 109| 73.50|62|66.40|121| 48.72|175| 14.16|109|54.99| 106  | **+0.93** |
>     | 50| 67.55 | 141   | 68.35   | 127| 73.92|105|68.32|143| 46.79|190| 14.24|117| 54.86| 137  | +0.86 |
>
>     [1] Beyond the 80/20 Rule: High-Entropy Minority Tokens Drive Effective Reinforcement Learning for LLM Reasoning.

---

> ### Author Response · Authors · 2025-11-22
> **Part3**
>
> - **Question 2 - Convergence and its influence of staged training**: In the first stage, **convergence is defined by the validation accuracy plateau (<5%)**. Since all of our comparisons with the general adaptive reasoning algorithms are based on the model after this first-stage GRPO training, any impact from this stage is consistent across experiments, ensuring a fair comparison. The first stage training is sufficiently converged; if it were not, the performance of the resulting reasoning model would not be maximized. As a result, the second stage performance would be lower, and the response length would be longer (because the model’s capacity would still be insufficient, and the entire dataset would introduce additional difficulty, causing an increase in the average output length). We have included baseline results after 50 steps in the table below, which validate this point. **For adaptive calibration performance, there is no impact, as our method is based on the model’s true capacity**. The stronger the model, the greater the compression factor it can achieve, which is a straightforward observation.
>
>     | Model| PubMedQA Acc. | PubMedQA Len. | MMLU-ProM Acc. | MMLU-ProM Len. | GPQA-M Acc. | GPQA-M Len. | MedXpert Acc. | MedXpert Len. | AES    |
>     |---------------------|---------------|---------------|----------------|----------------|-------------|-------------|----------------|----------------|-------------|
>     | Qwen-GRPO (step 200) | 72.30    | 396 | 66.12| 535  | 48.46  | 575    | 14.48| 592  | - |
>     | Kimi (step 50 base)  | 75.00    | 236 | 66.71| 338  | 46.92  | 376    | 14.89| 330  | 0.42 |
>     | CosFn | 75.10    | 885 | 23.32| 998  | 32.82  | 1012   | 10.12| 1020 | -2.36|
>     | DAST  | 69.70    | 153 | 65.86| 205  | 50.51  | 466    | 15.59| 288  | 0.47 |
>     | ShortBetter| 71.70    | 260 | 65.86| 376  | 46.92  | 405    | 14.36| 387  | 0.26 |
>     | AdaThink-Med    | 73.50    | 62  | 66.45| 121  | 48.72  | 175    | 14.16| 88   | **0.81** |
>     |&nbsp;|
>     | Qwen-GRPO (step 50)  | 72.50    | 358 | 58.35| 506  | 45.21  | 525    | 13.36| 419  | - |
>     | Kimi (step 50 base)  | 71.30    | 235 | 56.25| 296  | 44.33  | 476    | 12.99| 373  | 0.12 |
>     | CosFn | 72.70    | 176 | 56.88| 840  | 42.98  | 729    | 12.50 | 758  | -0.50|
>     | DAST  | 71.30    | 287 | 57.24| 353  | 44.56  | 425    | 13.04| 336  | 0.14 |
>     | ShortBetter| 72.80    | 330 | 58.11| 289  | 45.01  | 383    | 13.11| 452  | 0.18 |
>     | AdaThink-Med    | 73.00| 85  | 57.45| 150  | 45.10   | 224    | 13.25| 135  | **0.65** |
>
> - **Question 3 - Reasoning Modes correlate with specific medical task types**: Thank you for your keen observation. Indeed, as you mentioned, the emergent **"non-thinking" mode is strongly associated with **basic medical fact QA**, which focuses on knowledge retrieval and does not require complex multi-step reasoning**. This is evident in the **PubMedQA** dataset, which emphasizes knowledge retrieval and exhibits a large number of non-reasoning modes. On the other hand, the **"thinking" mode is more closely associated with complex tasks such as **prognosis prediction** and multi-step decision-making**. This is supported by evidence from datasets like **GPQA-M**, **MMLU-ProM**, and **MedXpertQA**, which focus on multi-step reasoning and clinical multi-evidence tasks, where reasoning modes are consistently maintained.
>
>
> - **Question 4 - Characteristics of the retained 40% data**: Thank you for your thorough observation. Specifically, the 40% of retained data **exhibits a balanced difficulty distribution, covering multiple difficulty stages**. This subset **emphasizes the inclusion of higher difficulty samples that offer significant training gains (samples that, through reasoning chain growth and RFT, can lead to performance improvements)**, which drives the observed performance boost. Beyond the **AlphaMed19k** dataset, we also observed similar patterns in the entropy of the open-ended training dataset provided by **HuatuoGPT-O1**, further **demonstrating the generalizability** of the algorithm.

---

> ### Author Response · Authors · 2025-11-22
> **Part4**
>
> - **Question 5 - How does AdaThink-Med handle medical questions with high inherent uncertainty**: The question "What is the most likely diagnosis for a patient with non-specific symptoms?" **involves high uncertainty and requires complex, multi-step reasoning**. In this case, AdaThink-Med will allocate more tokens to process the question, engaging in a longer and more detailed thought process. On the other hand, questions with low uncertainty but rare answers are fundamentally simpler in terms of knowledge retrieval. Although such questions may involve a long tail of less common knowledge, the model's internal knowledge base operates on the principle that if the information has been encountered before, it will be known; if not, it will remain unknown. As a result, **AdaThink-Med will perform a brief thought process, focusing on retrieving the relevant information**. In summary, AdaThink-Med determines whether to engage in complex reasoning or a simpler retrieval process based on the inherent complexity of the question itself.
>
>
> - **Question 6 - Subpopulations Evaluation**: Thank you for your valuable suggestion. In fact, questions about pediatric vs. adult medicine are present in significant portions of the **PubMedQA** and **MedMCQA** datasets. Additionally, questions related to common vs. rare diseases are heavily represented in the **MedXpertQA** dataset, which focuses on clinical reasoning. We have conducted extensive validation on these benchmarks in the paper, and the results demonstrate the significant superiority of our algorithm.

---

> ### Comment · Reviewer_52Pp · 2025-11-25
>
> I believe the authors have solved most of my problems; I shall refer to the comments of other reviewers to maintain my positive score.
>
> Meanwhile, I noticed that the authors provided a lot of additional experiments and explanations during the rebuttal, and I hope that these revisions can be added to the updated version in future versions. Thanks.

---

> > ### Author Response · Authors · 2025-11-25
> >
> > Thank you for your positive feedback and for the time and effort you have dedicated to reviewing our work! We truly appreciate your recognition and supportive comments. As requested, we have incorporated the additional experiments and explanations into the updated manuscript (please refer to Section B.5 in the Appendix).
> >
> > Once again, we extend our sincere gratitude for your diligent efforts and valuable constructive feedback!

---

### Official Review · Reviewer_B55m · 2025-10-29

**Soundness:** 3
**Presentation:** 3
**Contribution:** 3
**Rating:** 4
**Confidence:** 4

**Summary:**

This paper proposes AdaThink-Med, a framework for medical large language models (LLMs) that enhances reasoning efficiency via uncertainty-guided adaptive length calibration. The key innovation is to use output entropy as an uncertainty measure to dynamically estimate the difficulty of each question, leveraging both correctness and uncertainty to guide the reasoning process. The method adaptively adjusts the reward for reasoning length during reinforcement learning: promoting concise reasoning for easy questions and allowing more extensive reasoning steps for difficult ones. Extensive experiments on multiple medical QA datasets demonstrate that AdaThink-Med significantly reduces token consumption without sacrificing accuracy, and in some cases improves reasoning performance.

**Strengths:**

1. The proposed uncertainty-guided adaptive length calibration is clearly formulated and technically well-motivated. The mechanism for estimating problem difficulty and dynamically adjusting output length is systematically constructed.
2. The work provides comprehensive empirical analysis, including comparisons to multiple baselines and ablation studies, offering evidence for the effectiveness and efficiency of the approach within the evaluated scope.

**Weaknesses:**

1. Although the experiments were conducted in the medical LLM domain, the proposed uncertainty-guided length calibration appears to be a generic method for reasoning efficiency, with limited explicit adaptation to medical-specific characteristics. It is unclear if the approach addresses any challenges unique to medical reasoning, or if it simply applies a general technique to medical datasets.
2. The paper demonstrates empirical improvements, but lacks discussion of the stability and reliability of the training procedure, especially with respect to reinforcement learning and the reward mechanism. There is insufficient analysis of potential failure cases or model sensitivity to hyperparameter choices and data distribution shifts.
3. The proposed framework relies on model-based uncertainty scores to define question difficulty, which may not align with human or domain-expert judgments. If there is substantial divergence between what the model perceives as "difficult" and what a human user finds challenging, the adaptive reasoning length strategy could become suboptimal or even counterproductive. The paper does not address how such misalignment could affect real-world applicability, nor suggests strategies to mitigate possible negative impacts when model and user difficulty perceptions differ.
4. The method is evaluated exclusively on medical QA datasets, but the technical design does not incorporate any medical-specific characteristics. This leads to a contradiction—whether the method is intended as a general reasoning efficiency approach or as a domain-specific optimization for medical LLMs. If domain-specific optimization is claimed, the manuscript should clarify which aspects of the approach are tailored to medical reasoning tasks.
5. All results are based on automatic metrics and benchmark datasets, with no involvement from medical professionals (e.g., clinicians, medical students) for qualitative or quantitative assessment. This lack of user-centered evaluation makes it difficult to ascertain the practical impact and real-world effectiveness of the method.
6. The analysis focuses primarily on efficiency and accuracy, but does not address potential side effects, such as generating unnecessarily verbose outputs for hard questions, or reducing token usage while missing important reasoning steps. A deeper exploration of these issues would help provide a more balanced evaluation of the approach.

**Questions:**

As indicated in the weaknesses section.

---

> ### Author Response · Authors · 2025-11-22
> **Part1**
>
> Thank you very much for taking the time to review and for your support. We try our best to address your questions as follows.
> - **Weakness 1 and Weakness 4 - Application Question**:
> Thank you for your thoughtful feedback. As you observed, **our method indeed can be used for general reasoning optimization but we foucus on medical domain**. The proposed entropy-based difficulty measure for adaptive reasoning **is novel both in the generic domain and in medical contexts and not simply the application of a general technique to medical datasets**. On one hand, this is because **we are working in the medical field, and on the other, the medical domain has a strong demand for clinically deployable solutions, which motivates us to address real-world deployment challenges**. In the medical field, there is an urgent need for deployment, with higher requirements for latency and inference costs, necessitating more efficient model services. **For example, in stroke scenarios, decisions regarding thrombolysis or surgery need to be made in a very short time because every second counts**. In such cases, the model must not only provide accurate results but also operate with high speed. **Our approach represents the first step in adaptive reasoning for medical applications**. We have conducted extensive validation using publicly available medical QA datasets, which demonstrates the effectiveness of our algorithm in addressing these challenges.

---

> ### Author Response · Authors · 2025-11-22
> **Part2**
>
> - **Weakness 2 Insufficient Analysis**: Thank you for your valuable suggestions. We would like to clarify the following three points.
>     1. **Insufficient anlysis of potential failure cases**: First, we have added an analysis of failure cases in Section 4.1 (Figure 5) of the revised manuscript.
>     2. **Model sensitivity to hyperparameters**: We have also included ablation experiments on the hyperparameters $K$ and temperature $T$ with the Qwen2.5-Instruct-7B model. The results show that **when $T = 1.0$ is fixed, changes in the $ \text{TOP-K} $ value lead to stable efficiency improvements in AdaThink-Med**. However, both very large and very small $ \text{TOP-K} $ values result in suboptimal efficiency improvements. The reason for this is that smaller $K$ values lead to fewer sampled tokens, causing inaccurate difficulty estimation, while larger $K$ values cause an averaging effect where the entropy of plain tokens smooths out the model's entropy. **When $ \text{TOP-K} = 20\% $, the impact of different temperatures on the final performance is minimal**. In our experiments, we found that the only noticeable difference was that higher temperatures accelerate the convergence speed. Based on our analysis and experimental results, AdaThink-Med demonstrates parameter stability, achieving optimal efficiency improvement with a larger temperature setting (which estimates a more diverse range of responses) and an appropriate token count (20%).
>         | TOPK | T | Med Acc. | Med Len. | MedMC Acc. | MedMC Len. | Pub Acc. | Pub Len. | MMLU Acc. | MMLU Len. | GPQA Acc. | GPQA Len. | MedX Acc. | MedX Len. | Avg Acc. | Avg Len. | AES|
>         |------|------|------------|------------|--------------|--------------|---------------|---------------|----------------|----------------|-------------|-------------|----------------|----------------|-----------|-----------|-------|
>         |-|-|63.86|490|65.68|359|73.10|386|62.23|523|46.41|562|12.28|585|52.44|484|-|
>         |5|1.0|67.83|102|67.53|125|74.22| 87| 65.93|131|46.96|221|13.95|102|54.41|128| +0.85 |
>         |20|1.0|68.34|79|68.74|109|73.50|62|66.40|121|48.72|175|14.16|109| 54.99| 106  | +0.93 |
>         |50|1.0| 67.55 | 141| 68.35| 127| 73.92 | 105 | 68.32| 143|46.79|190 |14.24|117|54.86|137|+0.86|
>         |20|0.3  | 68.05 | 83 | 69.03| 112| 73.99 | 96  | 65.97| 133  | 48.58  | 187|14.26| 96| 54.98| 118  | +0.90 |
>         |20|0.5  | 68.22 | 76 | 68.53| 121| 74.1| 87  | 66.21| 129| 47.96|169|14.10|93| 54.86| 113  | +0.91 |
>         |20|1.0 | 68.34 | 79 | 68.74| 109| 73.50 | 62| 66.40| 121|48.72|175 | 14.16| 109  | 54.99| 106  | +0.93 |
>
>
>     3. **Model sensitivity to data distribution shifts**: Additionally, we conducted experiments on new datasets and architectures. Specifically, we applied adaptive reasoning with **a new open-ended dataset** on **HuatuoGPT-O1**. The results show that even when the model and dataset are changed, i.e., when there is a **data distribution shift**, AdaThink-Med still demonstrates SOTA performance, showing its robustness.
>
>
>         | Model | Med Acc. | Med Len. | MedMC Acc. | MedMC Len. | Pub Acc. | Pub Len. | MMLU Acc. | MMLU Len. | GPQA Acc. | GPQA Len. | MedX Acc. | MedX Len. | Avg Acc. | Avg Len. | AES |
>         |---------------------------|------------|------------|--------------|--------------|---------------|---------------|----------------|----------------|-------------|-------------|----------------|----------------|----------|----------|------|
>         | HuatuoGPT-o1-8B[cot] | 76.35 | 568| 62.82| 445| 79.80 | 445 | 63.71| 521  | 54.35  | 585 | 17.06| 601  | 59.02 | 527 | - |
>         | Kimi| 71.01 | 373| 62.87| 302|77.40|288|67.04|340|50.26|401|14.49| 363  | 57.18 | 345 | 0.19 |
>         | CosFn  | 68.42 | 260| 62.63| 126| 77.30 | 159 | 65.60| 220  | 48.97  | 187 | 14.24| 246  | 56.20 | 200 | 0.38 |
>         | DAST| 71.17 | 730| 61.01| 619| 76.70 | 578 | 64.36| 697  | 45.90  | 794 | 15.02| 765  | 55.69 | 697 | -0.60|
>         | ShortBetter | 71.64 | 336| 63.18| 271| 78.40 | 258 | 66.12| 299  | 49.74  | 353 | 14.45| 321  | 57.26 | 306 | 0.27 |
>         | AdaThink-Med | 72.74 | 183| 63.54| 192| 76.60 | 72  | 66.91| 204  | 50.77  | 344 | 13.88| 221  | 57.41 | 203 | **0.48** |
>         |&nbsp;|
>         | m1-7B[cot]  | 75.01 | 2161  | 62.32| 1749 | 74.20 | 1080| 68.07| 2564 | 51.53  | 3887| 18.28| 3265 | 58.24 | 2451| - |
>         | Kimi| 71.23 | 1425  | 61.26| 992| 72.50 | 856 | 65.56| 1335 | 50.58  | 2069| 15.53| 1974 | 56.11 | 1442| 0.23 |
>         | CosFn  | 72.58 | 1566  | 60.57| 1011 | 72.70 | 774 | 67.99| 1218 | 51.35  | 1655| 16.23| 1563 | 56.90 | 1298| 0.36 |
>         | DAST| 70.87 | 2445  | 58.43| 1995 | 70.10 | 1243| 65.33| 2655 | 48.33  | 4096| 14.43| 4096 | 54.58 | 2755| -0.44|
>         | ShortBetter | 73.55 | 1359  | 60.58| 927| 71.40 | 839 | 65.37| 1290 | 50.12  | 1847| 14.99| 1885 | 56.00 | 1358| 0.25 |
>         | AdaThink-Med| 74.15 | 982| 63.35| 876| 73.10 | 585| 67.99| 1127 | 52.03  | 1596| 17.55| 1433| 58.03 | 1100| **0.53** |

---

> ### Author Response · Authors · 2025-11-22
> **Part3**
>
> - **Weakness 3 and Weakness 5 - Divergence with Model and Human, User-centered Evaluation**:
>
>     **1. Divergence with Model and Human**: Thank you for your insightful observation. In fact, **there are two directions for improving adaptive reasoning: one relies on the model's own (model-based) assessment of question difficulty, and the other incorporates human (human-based) preferences regarding difficulty**. The latter **requires collecting a large amount of human-labeled preference data**, such as the preference for response length for the same question. **Our focus is on model-based adaptive reasoning optimization, as it is scalable on a large scale**. After this stage, we can significantly reduce our reliance on human preference data, requiring only a small amount of human preference data for RFT (Reinforcement Fine-Tuning) to align with user difficulty perceptions. **Aligning with human preferences is part of future work, and our paper primarily explores the scalability and generalization of model-based adaptive reasoning optimization**, laying a solid foundation for incorporating human preferences in the future.
>
>     **2. User-centered Evaluation**: We agree that the model-based adaptive optimization algorithm should be tested in real-world applications. To address this, **we have added a user-centered evaluation**. Specifically, **we invited three clinicians to perform an algorithmic comparison on 500 randomly sampled samples from the test set**, focusing on the **accuracy of responses and the acceptability of reasoning redundancy**. Our AdaThink-Med received significantly more acceptance than other methods, demonstrating its superior generation quality as perceived by human evaluators.
>
>     | Method   | Response Accuracy (↑) | Redundancy Tolerance (↑) |
>     |---------------|-------------------|----------------------|
>     | Kimi| 71.25%  | 73%   |
>     | CosFn    | 70.34%  | 67%   |
>     | DAST| 68.37%  | 65%   |
>     | ShortBetter   | 70.23%  | 72%   |
>     | AdaThink-Med  | **76.25%**  | **89%**   |
>
>
> - **Question 6 - Potential Side Effects**: Thank you for your insightful feedback. We have conducted a deeper exploration of the following two aspects:
>
>     1. **Whether unnecessarily verbose outputs are generated for hard questions**: After training the base model with GRPO, we performed multiple response samplings for each question in the validation set and identified the bottom 20% of samples with the lowest average accuracy as "hard questions." After adaptive training, we compared the changes in output length for this subset, as shown in the table below. The results indicate that **AdaThink-Med does not generate unnecessarily verbose outputs for hard questions**, validating the effectiveness of our approach.
>
>         | Method    | MedQA Acc. | MedQA Len. | MedMCQA Acc. | MedMCQA Len. | PubMedQA Acc. | PubMedQA Len. | MMLU-ProM Acc. | MMLU-ProM Len. | GPQA-M Acc. | GPQA-M Len. | MedXpert Acc. | MedXpert Len. | Avg. Acc. | Avg. Len. |
>         |---------------|------------|------------|--------------|--------------|---------------|---------------|----------------|----------------|-------------|-------------|----------------|----------------|-----------|-----------|
>         | Qwen-GRPO| 56.13 | 876   | 52.68   | 683| 63.90    | 125 | 61.83| 674  | 43.27  | 890    | 13.15| 1033 | 48.49| 714  |
>         | AdaThink-Med  | **58.22** | **129**   | **54.25**   | **253**| **65.60**    | **76**  | **63.97**| **236**  | **46.53**  | **323**    | **14.28**| **567**  | **50.48**| **264**  |
>
>
>     2. **Whether important reasoning steps are missed in the process of reducing token usage**: Significant token reduction mainly occurs for easier questions. **Since key reasoning steps are difficult to quantify, we conducted a **User-centered Evaluation** with clinicians to address this**. As shown in the table below, **AdaThink-Med demonstrates SOTA response accuracy redundancy tolerance, indicating that overall we did not miss important reasoning steps in our efforts to reduce token usage**. This can also be explained by our training mechanism: if important reasoning steps were skipped to reduce token usage, the model's entropy would inevitably increase, which would lead to higher uncertainty. As a result, the difficulty would rise, causing the model to increase output length. Therefore, we did not reduce token usage at the cost of missing important reasoning steps.
>
>
>         | Method   | Response Accuracy (↑)  | Redundancy Tolerance (↑) |
>         |---------------|-------------------|----------------------|
>         | Kimi| 71.25%  | 73%   |
>         | CosFn    | 70.34%  | 67%   |
>         | DAST| 68.37%  | 65%   |
>         | ShortBetter   | 70.23%  | 72%   |
>         | AdaThink-Med  | **76.25%**  | **89%**   |

---

### Official Review · Reviewer_KxCJ · 2025-11-01

**Soundness:** 3
**Presentation:** 3
**Contribution:** 2
**Rating:** 4
**Confidence:** 4

**Summary:**

This paper proposes AdaThink‑Med, an end‑to‑end reinforcement‑learning framework that teaches medical LLMs to adapt the amount of reasoning to the difficulty of a question. The key idea is to estimate question difficulty with uncertainty, then shape a length reward that (i) penalizes unnecessarily long chains when the model is confidently correct on easy items and (ii) compensates by encouraging longer chains when the model is wrong on hard items.

1. Uncertainty is computed as the mean token entropy over the top‑K most uncertain tokens in a rollout, then min‑max normalized within the batch. Multiple rollouts per question are used.
2. These uncertainties discount a correctness‑based estimate of difficulty (D_q): correct-but‑uncertain generations make the item harder.
3. A difficulty‑aware length rewarduses a batch quantile threshold ( \theta_B ) to split easy vs. hard questions and applies asymmetric shaping: penalize long outputs for easy+correct cases; encourage longer outputs for hardincorrect cases. Final reward mixes accuracy, format, and length terms.
4. Staged training: first learn to reason via GRPO without length calibration, then enable adaptive length calibration with EMA‑smoothed batch statistics.

Results. On six medical QA benchmarks (MedQA, MedMCQA, PubMedQA, MMLU‑ProM, GPQA‑M, MedXpert), AdaThink‑Med reduces output length dramatically with minimal accuracy loss, achieving state‑of‑the‑art Accuracy‑Efficiency Score (AES) among length‑calibration methods. For example, with LLaMA‑3.1‑8B backbones, average length drops from 410 to 64 tokens  with only ~1.1% average accuracy decrease. Similar gains hold for Qwen‑2.5‑7B. The method also reports:

1. "Non‑thinking" vs "thinking" modes emerging spontaneously (short direct answers vs. concise reasoning).
2. Reward‑hack analysis: greedy length penalties can collapse both length and accuracy; the proposed performance compensation stabilizes training.
3. Ablations on the difficulty threshold ( \tau ) and uncertainty weight ( \alpha ), and a staged‑training study.
4. A dataset‑selection use‑case: selecting subsets by model‑induced output lengths; 40% subsets sometimes surpass full‑set training.

**Strengths:**

1. Principled difficulty signal: Combines correctness with trajectory‑level entropy to reflect confident‑correct vs. uncertain‑correct generations.
2. Asymmetric length shaping with performance compensation demonstrably prevents collapse seen in greedy penalties.
3. Substantial efficiency gains with small accuracy changes across six medical QA tasks and two backbones: e.g., LLaMA average 64 tokens vs 410 baseline; Qwen 106 vs 497.
4. Beats strong length‑calibration baselines in AES on both backbones.
5. Emergent "non‑thinking / thinking" modes without extra supervision.
6. Generalization to a different reasoning setup (HuatuoGPT‑o1) with sizable length reduction and small average accuracy decrease.

**Weaknesses:**

1. AES sensitivity & cost realism. AES uses fixed weights ( (\alpha,\beta,\gamma)=(1,3,5) ) that strongly favor length reductions (Table 3 even shows ( \tau=0.9 ) yielding extremely short outputs with the highest AES despite accuracy drop). No sensitivity analysis is provided; wall‑clock latency and decoder FLOPs are not reported. Token counts alone can misrepresent real cost (e.g., different backbones/tokenizers, batching, KV‑cache).
2. Difficulty estimation depends on batch min–max and quantile thresholding. While EMA smoothing is used, batch‑wise normalization can introduce instability or dependence on batch composition; an ablation on batch size / normalization would help.
3. Inference‑time mechanism is implicit. The text says AdaThink‑Med "estimates the difficulty of incoming questions" at inference, but there is no explicit gating; the policy implicitly emits shorter/longer chains. Make this explicit and clarify whether any multi‑sample inference is needed (it appears not, but this is important for deployability).
4. Dataset‑selection experiment lacks controls. The subset selection is driven by model‑induced output length (global median split), but there is no comparison to random or difficulty‑agnostic subsets at equal sizes; thus, it is unclear whether improvements over the full set arise from better data or regularization via less data.
5. Evaluation scope. All benchmarks are QA; clinical free‑text and multi‑step decision support are not assessed. Also, accuracy extraction in the open‑ended RL setup (no options) needs more detail—how "final answer" is parsed/validated across datasets.
6. Minor inconsistency. In "Comparison with length‑calibration methods," the text claims AdaThink reduces lengths "6.4x ... than Kimi 1.5," but Table 2 shows 64 vs 125 tokens  on LLaMA; the 6.4x ratio matches GRPO (410/64), not Kimi1.5. Please correct.
7. Tokenization comparability. Length comparisons mix Qwen and LLaMA tokenizers and multiple models. While AES baselines are set per‑backbone in Table 2, cross‑model averages in Table 1 may still reflect tokenizer effects. Clarify whether lengths are normalized or reported per‑model tokenizer.

**Questions:**

1. AES sensitivity / latency. How robust are conclusions to different AES weights? Please report wall‑clock decode latency and GPU‑FLOPs/token (or throughput) to corroborate the token‑length savings.
2. Inference‑time behavior. Does inference use a single sample per question with a learned stopping behavior, or any explicit difficulty predictor / early‑exit? If purely implicit, can you add a plot of length vs. estimated difficulty without multi‑sampling at test time?
3. Accuracy parsing in open‑ended RL. How is the final answer extracted when options are not presented? Is there a verifier or regex per dataset? Please detail the format reward and answer‑matching logic used in training and evaluation.
4. Batch normalization choices. How sensitive is performance to the min–max normalization and the quantile ( \tau ) under different batch sizes? Could per‑sample entropy calibration (e.g., running averages) remove batch coupling?
5. Top‑K entropy. What is (K) (or the fraction) used? Please include an ablation on K (and temperature (T)) to show robustness of the uncertainty estimator.
6. Dataset selection controls. Can you compare to random subsets and loss‑based / gradient‑based selection at the same sizes? Also specify the compute budget and whether training steps were matched across subset sizes.
7. Cross‑tokenizer fairness. For Table 1 averages, did you normalize lengths across tokenizers (e.g., bytes or characters) or compute per‑model averages only? A clarification would help interpret cross‑model length gaps.
8. Clinical realism. Any preliminary results on free‑text clinical tasks (e.g., short case vignettes) to test whether the "non‑thinking / thinking" mode switch remains beneficial beyond multiple‑choice QA?
9. Since mainstream SOTA LLMs do not support token precision thinking effort controling, I am unaware of the importance of doing budget controllon for madical domain. Please ellaborate this.

---

> ### Author Response · Authors · 2025-11-22
> **Part1**
>
> Dear reviewer KxCJ, thank you for your review and valuable suggestions regarding our work. Below, please find our responses to your concerns.
> - **Weakness 1 and Question 1- Senesitivity of AES and Cost Realism**:
>     1) **Senesitivity of AES**:
>     Thank you for raising the concern about the sensitivity of the AES metric to hyperparameter choices. We fully understand and appreciate your point. **The AES metric is widely adopted in the adaptive reasoning community, as demonstrated in prior works like[1][2][3], and we have used the default parameters from these established studies without any modifications** specific to our method. The AES was introduced as a comprehensive measure to balance efficiency gains and performance drops, ensuring fair comparison with existing approaches.
>
>    In addition to using the standard AES configuration, **we have added experiments with different hyperparameter settings for AES**. The results consistently demonstrate the superiority of AdaThink-Med, showing substantial output length reductions with only minimal accuracy degradation across various configurations. This further validates the robustness and effectiveness of our approach, highlighting its overall advantage in adaptive reasoning tasks.
>
>    | Model   | avg_acc | avg_len | delta_len | delta_acc | α = 1, β = 3, γ = 5 | α = 1, β = 5, γ = 5 | α = 1, β = 2, γ = 3 | α = 1, β = 2, γ = 10 | α = 0.5, β = 2, γ = 1 | α = 0.1, β = 1, γ = 2 |
>    |-----------------------------|---------|---------|-----------|-----------|---------------------|---------------------|---------------------|-----------------------|-----------------------|-----------------------|
>    | Llama-3.1-8B-Instruct[cot]  | 54.25   | 425.27  | -    | -    | -    | -    | -    | - | - | - |
>    | Llama-3.1-8B-Instruct  | 48.93   | 254.38  | 0.40 | -0.10 | -0.09    | -0.09    | 0.11    | -0.58 | 0.10| -0.16 |
>    | Llama-3.1-8B-Instruct[grpo] | 56.72   | 409.88  | 0.04 | 0.05 | 0.17    | 0.26   | 0.13  | 0.17| 0.11 | 0.05  |
>    | Kimi| 55.65   | 124.89  | 0.71 | 0.03 | 0.78    | 0.84    | 0.76    | 0.78| 0.40   | 0.10|
>    | CosFn    | 38.10   | 973.64  | -1.29    | -0.30    | -2.78    | -2.78    | -2.18    | -4.27 | -0.94| -0.72 |
>    | DAST  | 52.71   | 51.37   | 0.88 | -0.03| 0.74    | 0.74    | 0.79    | 0.60 | 0.41   | 0.03|
>    | ShortBetter   | 55.94   | 167.38  | 0.61 | 0.03 | 0.70    | 0.76    | 0.67    | 0.70 | 0.37 | 0.09 |
>    | AdaThink-Med-Llama | 55.59   | 63.81   | 0.85 | 0.02 | **0.92**    | **0.97**    | **0.90**    | **0.92** | **0.47** | **0.11**|
>    |&nbsp;|
>    | Qwen-Instruct-7B[cot]   | 52.44   | 484.08  | -    | -    | -    | -   | -    | -| -    | -    |
>    | Qwen-Instruct-7B   | 48.04   | 278.24  | 0.43 | -0.08| 0.01    | 0.01    | 0.17    | -0.41 | 0.13 | -0.13 |
>    | Qwen-Instruct-7B[grpo]  | 54.74   | 496.62  | -0.03 | 0.04 | 0.11    | 0.19    | 0.06    | 0.11 | 0.07 | 0.04   |
>    | Kimi| 55.41   | 298.09  | 0.38 | 0.06 | 0.55    | 0.67    | 0.50    | 0.55    | 0.31 | 0.10|
>    | CosFn    | 38.10   | 973.64  | -1.01| -0.27    | -2.38    | -2.38    | -1.83    | -3.75 | -0.78 | -0.65 |
>    | DAST| 54.54   | 283.01  | 0.42 | 0.04 | 0.54    | 0.62    | 0.50    | 0.54 | 0.29    | 0.08 |
>    | ShortBetter   | 54.28   | 335.19  | 0.31 | 0.04 | 0.41    | 0.48    | 0.38    | 0.41 | 0.22 | 0.07 |
>    | AdaThink-Med-Qwen  | 54.99   | 105.71  | 0.78 | 0.05 | **0.93**    | **1.02**    | **0.88**    | **0.93** | **0.49** | **0.13** |
>
>    [1]Deepscaler: Surpassing o1-preview with a 1.5b model by scaling rl, 2025b
>
>    [2]ShorterBetter: Guiding Reasoning Models to Find Optimal Inference Length for Efficient Reasoning
>
>    [3]O1-Pruner: Length-Harmonizing Fine-Tuning for O1-Like Reasoning Pruning
>
>     2) **Cost Realism**:  In our setup, we used the vLLM inference framework, where **for models with identical architecture, factors like batching and KV-cache management are held constant, making output length a linear proxy for cost under comparable conditions**. **For different architectures or tokenizers, we fully agree that wall-clock latency and decoder FLOPs can vary non-linearly with output length**. **To address this, we have separately reported results for Llama and Qwen backbones in Table 1(see the revised manuscript)**, we have further improved the presentation to ensure clear and comparable cost evaluations based on token counts, aligning with common practice in adaptive reasoning research. We believe these adjustments provide a transparent and realistic assessment of the efficiency gains achieved by AdaThink-Med.

---

> ### Author Response · Authors · 2025-11-22
> **Part2**
>
> - **Weakness 2 and Question 4 - Ablation on Batch Size and Normalization**: Thank you for your insightful and valuable question. We would like to clarify that the introduction of Batch Normalization (BN) serves two main purposes. First, it **constrains the output uncertainty within the [0, 1] range, ensuring the stability of the algorithm**. Second, it **works in conjunction with EMA (Exponential Moving Average) to estimate the overall difficulty threshold of the dataset**. The overall difficulty of the dataset determines the extent to which adaptive reasoning can achieve compression—simpler datasets allow for greater compression. Therefore, combining BN with the EMA difficulty threshold helps estimate the overall difficulty of the training set. If a sample’s difficulty is below the overall threshold and the response is correct, a length penalty is applied to reduce redundant reasoning.
>     1. **Ablation of BN and Batch Size**. As per your suggestion, we have added ablation experiments on batch size and the presence of BN, as shown in the table below. The results indicate that **larger batch sizes lead to more accurate difficulty estimations, thereby accelerating the convergence of the algorithm. Smaller batch sizes take longer to converge but eventually achieve comparable results**. However, **when batch normalization is removed**, the absence of fixed value ranges and the lack of comparison with other samples (which BN achieves through normalization) **causes the adaptive reasoning algorithm to fail**. As a result, the model is unable to estimate the correct difficulty threshold, leading to training instability, and the final performance is nearly identical to that of the initial model.
>     2. **Ablation of the quantile $\tau$ under different batch size, per-sample entropy calibration**: When the threshold $\tau$ is fixed, the results in the table below show that the **batch size primarily affects the speed of convergence, with minimal impact on final performance**. In our experiments with **per-sample entropy calibration, we observed very slow convergence and instability during training, which is why we have not included those results**. In fact, per-sample entropy corresponds to using a batch size of 1 for overall difficulty estimation, which is both slow and inaccurate. In contrast, our approach, which combines batch normalization and EMA for overall difficulty estimation, ensures stable training and faster convergence.
>
>    | Model   | BN  | Batch Size | Med Acc. | Med Len. | MedMC Acc. | MedMC Len. | Pub Acc. | Pub Len. | MMLU Len. | MMLU Acc. | GPQA Len. | GPQA Acc. | MedX Len. | MedX Acc. | Avg Acc. | Avg Len. | Converge Step | AES   |
>    |-----------------------------|------|------------|------------|------------|--------------|--------------|---------------|---------------|----------------|----------------|-------------|-------------|----------------|----------------|----------|----------|---------------|--------|
>    | Llama-3.1-8B-Instruct[cot]  | -    | -| 68.18 | 461   | 57.32   | 309| 78.20    | 296 | 60.06| 457  | 45.64  | 485    | 16.08| 545  | 54.25    | 425 | -   | - |
>    | Llama-3.1-8B-Instruct  | -    | -| 54.75 | 302   | 54.69   | 169| 77.00    | 113 | 57.00| 256  | 35.64  | 345    | 14.48| 342  | 48.93    | 254 | -   | -0.09 |
>    | Llama-3.1-8B-Instruct[grpo] | -    | -| 72.50 | 438   | 61.60   | 299| 78.50    | 290 | 63.51| 442  | 46.92  | 461    | 17.26| 529  | 56.72    | 410 | -   | +0.17 |
>    | AdaThink-Med-ablations(BN)  | x    | 256x8 | 70.43 | 425   | 59.33   | 267| 78.20    | 275 | 62.55| 429   | 46.49  | 423    | 17.00| 500  | 55.76    | 387 | Fluctuation   | +0.17 |
>    | AdaThink-Med-ablations(bs)  | √    | 32x8  | 66.98 | 40    | 60.21   | 82 | 78.90    | 60  | 62.65| 93   | 46.53  | 96    | 15.76| 42   | 55.04    | 69  | 280  | +0.88 |
>    | AdaThink-Med  | √    | 256x8 | 67.00 | 36    | 60.41   | 73 | 78.40    | 54  | 63.58| 85   | 47.94  | 100    | 16.20| 35| 55.59    | 64  | 200   | +0.92 |

---

> ### Author Response · Authors · 2025-11-22
> **Part3**
>
> - **Weakness 3 and Question 2- Inference-time Mechanism**:Thank you for this insightful question regarding the inference-time mechanism of AdaThink-Med. You are correct in noting that **there is no explicit gating mechanism or network during inference. This is actually a key advantage of our method**. After the adaptive training process, which incorporates uncertainty-guided difficulty estimation, **the model’s is directly optimized to autonomously and implicitly determine the appropriate reasoning length for each input question**. This means the model learns to assess the difficulty of a problem and self-regulate the length of its output accordingly, without requiring an explicit gating mechanism or additional entropy calculation during inference.
>
>     Furthermore, we would like to clarify that **no multi-sample inference is needed during deployment**. **Multi-sample inference is only utilized during training for difficulty estimation**. Once the model is trained, the deployed version is highly efficient and straightforward for real-world applications, as it requires only a single forward pass per question to produce its adaptively scaled output. **In this regard, the mechanism is similar to how GPT-5 adapts its response length based on input question difficulty**, without the need for manual gating switches like in models such as Qwen-3. This approach ensures that AdaThink-Med is both highly efficient and deployable in practical scenarios.
>
>     Additionally, **we have included the requested plot of length vs. estimated difficulty in Figure 6 of the revised manuscript**. The overall trend observed is as follows: for correct test samples, as the response length increases, the uncertainty also rises, indicating a higher level of difficulty. For incorrect samples, both short and long response lengths show high difficulty, with short responses exhibiting a more pronounced increase, suggesting that more reasoning or thinking is required for such cases.
>
> - **Weakness 4 and Question 6- Dataset‑selection Experiment**: Thank you for your valuable suggestion. In response, we have **added experiments with randomly selected subsets as controls for each ratio**. To ensure a fair comparison, **we specifically matched the compute budget (training steps = 200) across different subset sizes, allowing the model to fully converge at each ratio**. The results clearly demonstrate that our data selection method outperforms random subset selection, **confirming that the improvements stem from better training data, not simply from reducing the dataset size**.
>
>     | **$\delta$** | **Method** | **MedQA Acc.** | **MedQA Len.** | **MedMCQA Acc.** | **MedMCQA Len.** | **PubMedQA Acc.** | **PubMedQA Len.** | **MMLU-ProM Acc.** | **MMLU-ProM Len.** | **GPQA-M Acc.** | **GPQA-M Len.** | **MedXpert Acc.** | **MedXpert Len.** | **Avg. Acc.** | **Avg. Len.** |
>     |--------------|------------|----------------|----------------|------------------|------------------|-------------------|-------------------|--------------------|--------------------|-----------------|-----------------|-------------------|-------------------|----------------|----------------|
>     | **0%**  | -| 63.86| 490  | 56.68  | 359    | 73.10   | 386| 62.28    | 523 | 46.41 | 562   | 12.28   | 585| 52.44| 484  |
>     | **20%** | random| 64.21| 505  | 57.23  | 384    | 73.50   | 391| 62.88    | 537 | 46.99 | 588   | 13.22   | 577| 53.01| 497  |
>     | **20%** | ours  | 64.81| 497  | 58.50  | 373    | 74.40   | 400| 64.82    | 526 | 48.21 | 565   | 14.00   | 585| 54.12| 491  |
>     | **40%** | random| 64.38| 523  | 57.66  | 358    | 72.30   | 415| 66.65    | 515 | 50.22 | 557   | 13.99   | 585| 54.20| 493  |
>     | **40%** | ours  | 65.44| 512  | 58.81  | 367    | 74.30   | 396| 66.51    | 522| 51.28 | 564   | 14.37   | 590| 55.12| 492  |
>     | **60%** | random| 64.96| 507  | 57.43  | 352    | 72.60   | 416| 65.38    | 532 | 48.32 | 532   | 14.02   | 551| 53.79| 482  |
>     | **60%** | ours  | 68.50| 488  | 58.69  | 356    | 73.10   | 381| 66.12    | 513 | 47.44 | 535   | 13.96   | 569| 54.64| 474  |
>     | **80%** | random| 66.38| 499  | 56.73  | 366    | 71.20   | 398| 66.33    | 515 | 48.96 | 528   | 14.67   | 560| 54.05| 478  |
>     | **80%** | ours  | 68.42| 504  | 58.38  | 370    | 72.30   | 388| 67.49    | 527 | 50.00 | 576   | 14.73   | 579| 55.22| 491  |
>     | **100%**| -| 67.55| 504  | 59.52  | 378    | 72.30   | 396| 66.12    | 535 | 48.46 | 575   | 14.48   | 592| 54.74| 497  |

---

> ### Author Response · Authors · 2025-11-22
> **Part4**
>
> - **Weakness 5 and Question 3 and Question8 - Evaluation Scope and Open-ended RL Setup**:
> 1. **Evaluation Scope**: Thank you for your careful observation and valuable suggestions. Regarding **clinical free-text generation**, we have added experimental results on the **radiology report generation task**. Specifically, we used the **MIMIC-CXR-2.0** dataset for validation. In the test set, we employed **LLM-as-judge** to evaluate the generated reports and compare them to the ground truth in two aspects: the completeness of the generated report and its conciseness. The results show that AdaThink-Med effectively compresses the report length while providing a complete report, demonstrating exceptional effectiveness.
>
>     | Method   | Completeness ↑ | Redundancy ↓ |
>     |---------------|----------------|--------------|
>     | Kimi| 66.2 | 42.5    |
>     | CosFn    | 34.5 | 54.1    |
>     | DAST| 46.7 | 50.7    |
>     | ShortBetter   | 55.3 | 45.2    |
>     | AdaThink-Med  | **78.4** | **22.5**    |
>
>
>    For **multi-step decision support**, the **MedXpertQA** dataset we used already contains a large number of such multi-step reasoning questions, which fully addresses this concern.
>
> 2. **Open-ended RL Setup**: We first adopted **HuatuoGPT-O1** as the base model and used the open-ended dataset provided in the paper. This dataset consists of open-ended QA pairs, where no answer options are provided in the question. For training and evaluation, we used the validation set from HuatuoGPT-O1. Specifically, **the accuracy verification for the open-ended RL setup is distilled from GPT-4o**. Considering the high correctness of GPT-4o but its expensive cost, we employed a distillation strategy to transfer GPT-4o’s evaluation capabilities to a smaller, open-source LLM. We filtered 20,000 data points from GPT-4o's output, resulting in a 3B verifier. **During training, both the generated content and the ground truth answer were fed into the verifier, which output a True/False response.**
>
>    The prompts used were as follows:
>
>     **Reasoning Prompt**:
>     ```text
>     "{QUESTION}\nPlease reason step by step, and put the final answer in \\boxed{{}}."
>     ```
>     We parsed the answer from within the `\\boxed{}` and verify its correctness using follow prompt:
>
>     **Verification Prompt**:
>     ```text
>         "<Model Response> {} </Model Response>
>         <Reference Answer> {} </Reference Answer>
>         Your task is to evaluate the model response by comparing it to the reference answer. If the model response is correct and aligns with the reference answer, output 'True'. If it is incorrect or fails, output 'False'."
>     ```
>
> - **Weakness 6 - Minor Inconsistency**: Thank you for pointing out this inconsistency in our length reduction comparison. We have carefully corrected this statement in the revised manuscript to accurately. We greatly appreciate your meticulous review in catching this discrepancy.
> - **Weakness 7 and Question 7 Tokenization comparability**: Thank you for pointing that out. For Table 1, we initially overlooked the effects of different tokenizers across models and only computed per-model averages. In fact, it would be more appropriate to separate the results based on the backbone model type for a fairer comparison. We have addressed this issue in the revised version, and the results are now presented separately, taking tokenizer effects into account.

---

> ### Author Response · Authors · 2025-11-22
> **Part5**
>
> - **Question 5 Top‑K entropy**: Thank you for your careful observation. The value of $K$ used in our paper is consistent with the proportion employed in the large model entropy research paper [1], specifically 20%. As suggested, we have included ablation experiments on $K$ and temperature $T$ with the Qwen2.5-Instruct-7B model. The results show that **when $T = 1.0$ is fixed, changes in the $\text{TOP-K}$ value lead to stable efficiency improvements in AdaThink-Med**. However, both very large and very small $\text{TOP-K}$ values result in suboptimal efficiency improvements. The reason for this is that smaller $K$ values lead to fewer sampled tokens, causing inaccurate difficulty estimation, while larger $K$ values cause an averaging effect where the entropy of plain tokens smooths out the model's entropy.
>
>     **When $\text{TOP-K} = 20\%$, the impact of different temperatures on the final performance is minimal**. In our experiments, we found that the only noticeable difference was that higher temperatures accelerate the convergence speed. Based on our analysis and experimental results, AdaThink-Med demonstrates parameter stability, achieving optimal efficiency improvement with a larger temperature setting (which estimates a more diverse range of responses) and an appropriate token count (20%).
>
>     | TOPK | T    | MedQA Acc. | MedQA Len. | MedMCQA Acc. | MedMCQA Len. | PubMedQA Acc. | PubMedQA Len. | MMLU-ProM Acc. | MMLU-ProM Len. | GPQA-M Acc. | GPQA-M Len. | MedXpert Acc. | MedXpert Len. | Avg. Acc. | Avg. Len. | AES   |
>     |------|------|------------|------------|--------------|--------------|---------------|---------------|----------------|----------------|-------------|-------------|----------------|----------------|-----------|-----------|-------|
>     | -    | -    | 63.86 | 490   | 56.68   | 359| 73.10    | 386 | 62.28| 523  | 46.41  | 562    | 12.28| 585  | 52.44| 484  | -|
>     | 5    | 1.0    | 67.83 | 102   | 57.53   | 125| 74.20    | 87  | 65.98| 131  | 46.96  | 221    | 13.95| 102  | 54.41| 128  | +0.85 |
>     | 20   | 1.0    | 68.34 | 79    | 58.74   | 109| 73.50    | 62  | 66.45| 121  | 48.72  | 175    | 14.16| 88 | 54.99| 106  | +0.93 |
>     | 50   | 1.0    | 67.55 | 141   | 58.35   | 127| 73.90    | 105 | 68.32| 143  | 46.79  | 190    | 14.24| 117  | 54.86| 137  | +0.85 |
>     | 20   | 0.3  | 68.05 | 83    | 59.03   | 112| 73.90    | 96  | 65.98| 133  | 48.58  | 187    | 14.26| 96   | 54.97| 118  | +0.90 |
>     | 20   | 0.5  | 68.22 | 76    | 58.53   | 121| 74.10| 87  | 66.23| 129  | 47.96  | 169    | 14.10 | 93   | 54.86| 113  | +0.91 |
>     | 20   | 1.0    | 68.34 | 79    | 58.74   | 109| 73.50    | 62  | 66.45| 121  | 48.72  | 175    | 14.16| 88 | 54.99| 106  | +0.93 |
>
>
> - **Question9 - Importance of  Budget Control for Medical Domain**: In general, adaptive reasoning aims to enhance the inference efficiency and intelligence boundary of large models, while also saving context in multi-agent environments. This has been explored in models like **Qwen3, DeepSeek3.1, GPT-5, and GPT-5.1**, which incorporate adaptive efficiency optimizations. **While current SOTA LLMs are focused on pushing the limits of performance, we should also continue striving for improvements in efficiency**. In the medical domain, there is an **urgent need for deployment due to the high requirements for latency and inference costs, demanding more efficient model services**. For example, in stroke scenarios, decisions regarding thrombolysis or surgery need to be made in a very short time, as every second counts. In such situations, the model must not only provide accurate results but also operate with high speed. **Our approach represents the first step in adaptive reasoning for medicine**. We have conducted extensive validation using publicly available medical QA datasets and further supplemented clinical validation with a clinical imaging report generation task. These results all demonstrate the effectiveness of our algorithm. In summary, our work in the medical field lays a solid foundation for developing highly intelligent and efficient medical agents and models that can be deployed in real-world applications. We believe that future Medical LLMs/agents will be both efficient and intelligent.

---

> ### Comment · Reviewer_KxCJ · 2025-11-24
>
> Thank you for your detailed response. The quantitative experiments have addressed my earlier concerns. However, I would still like to discuss the authenticity of the thinking traces. From the current presentation, I do not see clear evidence that controllability over thinking length improves the authenticity of these traces, which is particularly important for real-world medical applications where practitioners must evaluate the reliability of model reasoning. Additionally, while adaptive length may contribute to higher MCQA accuracy, it does not necessarily guarantee that the generated reasoning traces are correct or faithful.
>
> Finally, I would like to raise a question regarding methodological novelty. As it stands, the approach appears closely related to applying reinforcement learning to control sequence length, as in prior work [1]. With these considerations, I am inclined to increase my score to a 6, but I still view the paper as borderline.
>
> [1] Aggarwal, Pranjal, and Sean Welleck. “L1: Controlling how long a reasoning model thinks with reinforcement learning.” arXiv preprint arXiv:2503.04697 (2025).

---

> > ### Author Response · Authors · 2025-11-24
> >
> > Dear reviewer KxCJ, we greatly appreciate your constructive feedback and your recognition of our contributions. We hope our response below will further enhance your confidence in our work.
> > - **1. Authenticity of the Thinking Traces**: Thank you for your insightful comments and professional perspective. **The core assumption behind our method is that models need to engage in sufficient reasoning before arriving at the correct answer. The adaptive algorithm ensures that the maximum reduction in reasoning length corresponds to the model's required minimum reasoning threshold**. We fully agree with your concern regarding the authenticity of thinking traces, and we have added an additional experiment during the rebuttal phase to address this. Specifically, **we invited three clinicians to compare the algorithm's output on 500 randomly sampled test cases, focusing on both the accuracy of the responses and the acceptability of reasoning redundancy (i.e., whether the reasoning included a minimum amount of reliable and correct reasoning content)**. Our AdaThink-Med received significantly more votes for superior generation quality compared to other methods, demonstrating both correctness and fidelity in its reasoning traces, as perceived by human evaluators. We believe this experiment can provide clear evidence of the model's improved performance and reasoning authenticity, and we will include it in the revised version.
> >
> >     | Method   | Response Accuracy (↑) | Redundancy Tolerance (↑) |
> >     |---------------|-------------------|----------------------|
> >     | Kimi| 71.25%  | 73%   |
> >     | CosFn    | 70.34%  | 67%   |
> >     | DAST| 68.37%  | 65%   |
> >     | ShortBetter   | 70.23%  | 72%   |
> >     | AdaThink-Med  | **76.25%**  | **89%**   |
> >
> >
> >
> > - **2. Difference from L1 and RFT-Based Sequence Control Methods**: Thank you for raising this very important point. We originally have detailed the differences between our algorithm and these methods in the related work section of the paper. Specifically, previous approaches primarily rely on the accuracy of problem sampling to classify question difficulty and adjust output lengths to improve reasoning efficiency for simpler questions. **In contrast, our method is the first to explicitly consider the inherent uncertainty in the answers themselves. We suppress low-uncertainty responses for simple questions (which are genuinely easy for the model), while encouraging longer reasoning for complex questions where there is higher uncertainty to explore alternative solutions**. This is a key distinction from prior methods.
> >
> >
> >     Additionally, regarding a direct comparison with L1: **L1 employs a fixed range for maximum and minimum lengths (e.g., n_max=4000, n_min=100)**, and during training, **it randomly samples an optimal response length within this range, aiming to optimize reasoning efficiency by reducing average response lengths**. However, this approach can be highly heuristic and introduces significant randomness, as difficult questions clearly need longer responses, rather than a random selection of lengths within the specified range. **In contrast, our adaptive reasoning method offers much greater flexibility by determining the optimal length based on the difficulty of the question, not by hard-coding a random length**. Specifically, we define difficulty in terms of answer uncertainty, **applying different optimal target lengths for easy and difficult questions (see line 245 in our paper), which is a novel and problem-centric design**. This ensures that simple questions generate shorter answers, while wrong difficult questions have longer explorations, unlike L1, which randomly samples lengths within a fixed range. Overall, our method differs fundamentally from both RFT-based and L1 approaches. We introduce a novel entropy-based difficulty measure for questions and optimize reasoning efficiency by grouping questions based on difficulty (easy vs. hard).

---

> > > ### Comment · Reviewer_KxCJ · 2025-11-24
> > >
> > > I thank the authors for their comprehensive response and the significant effort put into running additional experiments.
> > >
> > > Regarding novelty, if we isolate the core technical idea in AdaThink-Med, it can be summarized as:
> > >
> > > 1. Using token-level / top-K entropy to build an uncertainty score for each trajectory.
> > > 2. Aggregating multi-sample correctness and uncertainty into a scalar difficulty score per question.
> > > 3. Using a batch quantile to label samples as easy vs. hard.
> > > 4. Plugging this into a length-aware reward inside GRPO:
> > >     1. easy + correct $\to$ penalize long chains;
> > >     2. hard + wrong $\to$ encourage longer chains (performance compensation).
> > >
> > > However, very similar concepts have already been proposed, and the space of "entropy to guide thinking length" is now fairly crowded:
> > >
> > > 1. SEED-GRPO [1]: also uses "semantic entropy $\to$ per-sample weight inside GRPO". Their entropy is semantic (multi-sample), but functionally it is still entropy-based uncertainty controlling RL. They do not explicitly tie it to length, but in reasoning tasks, higher entropy typically correlates with harder problems and longer CoTs.
> > > 2. SEAT / Adaptive Termination [2]: computes semantic entropy at each round and stops refinement once SE drops below a threshold. This is essentially "entropy-based difficulty/uncertainty $\to$ stop/continue reasoning".
> > > 3. Step Entropy Compression [3]: uses per-step entropy to decide which parts of the chain to prune and trains the model (via GRPO) to emit compressed CoTs. Here, entropy directly drives chain length through RL, albeit with a different objective (compression vs. difficulty-aware allocation).
> > > 4. Entropy-Guided Reasoning Compression [4]: explicitly frames length–accuracy trade-offs in entropy terms and separates compression and exploration phases based on entropy dynamics.
> > >
> > > Taken together, these works show that using entropy as an uncertainty/difficulty signal and then using that to control reasoning length or RL updates is now a well-established pattern.
> > >
> > > For AdaThink-Med, the proposed contributions are: 1) Particular difficulty formula: This is a specific design choice; it does not appear exactly in prior work, but conceptually it is still "entropy-weighted correctness". 2) Batch-quantile $\theta_B$ + EMA + two-stage training. 3) Accuracy-aware length reward. 4) Application to medical RL with a dataset-selection side-effect.
> > >
> > > Overall, I find the novelty somewhat limited, even though this is, to my knowledge, the first application of this family of ideas to the medical domain. A narrower and more accurate claim such as "We propose a specific entropy-based difficulty estimator and plug it into a difficulty-aware length reward inside GRPO, targeting medical reasoning models, and show emergent non-thinking vs. thinking modes plus dataset-selection benefits." would be more plausible.
> > >
> > > In addition, there is still no update on human evaluation; the response currently only reports quantitative results.
> > >
> > > That said, I appreciate the thoroughness of the experimental section and have increased my score accordingly.
> > >
> > > [1] Chen, Minghan, et al. "Seed-grpo: Semantic entropy enhanced grpo for uncertainty-aware policy optimization." arXiv preprint arXiv:2505.12346 (2025).
> > >
> > > [2] Xu, Zenan, et al. "Adaptive termination for multi-round parallel reasoning: An universal semantic entropy-guided framework." arXiv preprint arXiv:2507.06829 (2025).
> > >
> > > [3] Li, Zeju, et al. "Compressing chain-of-thought in llms via step entropy." arXiv preprint arXiv:2508.03346 (2025).
> > >
> > > [4] Zhu, Hourun, et al. "Entropy-Guided Reasoning Compression." arXiv preprint arXiv:2511.14258 (2025).

---

> > > > ### Author Response · Authors · 2025-11-25
> > > >
> > > > Thank you very much for your positive discussion and constructive feedback. We sincerely appreciate the time and effort you have dedicated to reviewing our work, as well as your highly professional summary of related research, which has helped us better enrich the content of our manuscript and clarify our contributions. We have incorporated the relevant discussions into the related work section of the revised manuscript to better highlight our contributions. Additionally, we have now included the human evaluation results in the revised version (see Section B.5 of the Appendix).
> > > >
> > > > Once again, we extend our sincere gratitude for your thorough review and insightful suggestions.

---

### Official Review · Reviewer_xhiU · 2025-11-04

**Soundness:** 3
**Presentation:** 2
**Contribution:** 1
**Rating:** 4
**Confidence:** 2

**Summary:**

The authors propose an end-to-end framework, AdaThink-Med, for curbing response length of medical LLMs without compromising accuracy. Roughly speaking, the framework encourages model to think less on "easier" questions and think longer on "hard" questions. Uncertainty over questions (measured via entropy) is used to shape the length reward during training and help determine reasoning effort (and therefore, length) at inference. The approach is tested over several medical benchmarks where AdaThink-Med scores the highest on the AES (Accuracy-Efficiency Score) metric.

**Strengths:**

- Strong motivation. Indeed, reasoning length is a severe disadvantage of thinking models. Developing a training paradigm for length calibrations based on entropy over candidate outputs in the medical domain appears to be novel.

- The approach achieves the best AES according to chosen hyper-parameters.

- 6.4x length reduction against baseline, scaling from hundreds of tokens to a few dozen.

- The emergent "no thinking" versus "thinking" mode as a result of length-calibration training is an interesting observation that could help influence similar training strategies.

**Weaknesses:**

- Perhaps the most concerning, the accuracy drop over several benchmarks are too large to justify decreased output length. Decreased performance by over 6 points across several benchmarks. If the authors were reducing responses from several thousand to only a few dozen, this drop off could be justified, but it's often from a few hundred tokens to just a few dozen.

- The entropy estimate is sensitive to choice of LLM. While acknowledged by the authors, it is unclear how deriving responses from mixtures or LLMs or a wider breadth of LLMs would affect this strategy.

- The AES seems very sensitive to choice of hyper-parameters, rendering the superiority of AdaThink-Med along this metric less interpretable.

- While "excessive length" is subjective, most competitors in Table 1 are generating only a few hundred tokens. Compression strategies tend to be concerned with lengths numbering in the tens of thousands, which is common for benchmarks such as MATH500 and AIME24 (the types of benchmarks for which it can be argued that thinking models were developed for).

**Questions:**

- See weaknesses.
 - Does this approach generalize to larger models?
 - Does this approach generalize to newer models? (i.e., Qwen-3)
 - What are the closed source baseline results? (i.e., GPT-5, Gemini-2.5-Pro, etc.)

---

> ### Author Response · Authors · 2025-11-22
> **Part1**
>
> Dear reviewer xhiU, thank you for your review and valuable suggestions regarding our work. Below, please find our responses to your concerns.
>
> - **Weakness 1 - Accuracy Drop**: Thank you for your thoughtful feedback. We appreciate the reviewer’s concern regarding accuracy drop. However, we believe there may be a factual misunderstanding on this point. We would like to clarify that our primary baselines are GRPO-Llama and GRPO-Qwen, as our approach first enhances reasoning via GRPO fine-tuning before applying length calibration for adaptive thinking. **The reported accuracy drop is not consistent across multiple benchmarks**; it is primarily observed in MedQA, where AdaThink-Med-Llama shows a decrease of about 5.5 points compared to GRPO-Llama. However, on other datasets like MedMCQA, PubMedQA, MMLU-ProM, GPQA-M, and MedXpertQA, the accuracy changes are minimal, with some even showing slight improvements. This demonstrates that our method effectively balances efficiency and performance, with major drops confined largely to one dataset, and we are actively investigating MedQA-specific adaptations to address this outlier.
>
>     | Model| MedQA Acc. | MedQA Len. | MedMCQA Acc. | MedMCQA Len. | PubMedQA Acc. | PubMedQA Len. | MMLU-ProM Acc. | MMLU-ProM Len. | GPQA-M Acc. | GPQA-M Len. | MedXpert Acc. | MedXpert Len. | Avg. Acc. | Avg. Len. |
>     |---------------------|------------|------------|--------------|--------------|---------------|---------------|----------------|----------------|-------------|-------------|----------------|----------------|-----------|-----------|
>     | Llama-3.1-8B-Instruct[cot] | 68.18 | 461   | 57.32   | 309| 78.20    | 296 | 60.06| 457  | 45.64  | 485    | 16.08  | 545 | 54.25  | 425  |
>     | Llama-3.1-8B-Instruct    | 54.75 | 302   | 54.69   | 169| 77.00    | 113 | 57.00| 256  | 35.64  | 345    | 14.48| 342  | 48.93| 254  |
>     | Llama-3.1-8B-Instruct[grpo] | 72.50| 438   | 61.60   | 299| 78.50    | 290 | 63.51| 442  | 46.92  | 461    | 17.26| 529  | 56.72| 410  |
>     | AdaThink-Med-Llama   | **67.00(-5.50)** | 36    | **60.41(-1.19)**   | 73 | **78.40(-0.10)**    | 54  | **63.58(+0.07)**| 85  | **47.94(+1.02)**  | 100    | **16.20(-1.06)**| 35   | **55.59(-1.13)**    | 64   |
>     | Kimi | 70.62 | 145   | 60.93   | 100| 77.50    | 92  | 63.71| 164  | 45.38  | 118    | 15.75| 131  | 55.65| 125  |
>     | CosFn| 37.94 | 1007  | 49.29   | 919| 75.10    | 885 | 23.32| 998  | 32.82  | 1012    | 10.12| 1020 | 38.10| 974  |
>     | DAST | 60.64 | 30    | 56.90   | 34 | 76.20    | 39  | 59.48| 71  | 47.69  | 53    | 15.35    | 82 | 52.71| 51  |
>     | ShortBetter| 71.48 | 175   | 60.62   | 134| 78.60    | 118 | 62.99| 188  | 45.89  | 219    | 16.08| 171  | 55.94| 167  |
>     |&nbsp;|
>     | Qwen-Instruct-7B[cot]| 63.86 | 490   | 56.68   | 359| 73.10    | 386 | 62.28| 523  | 46.41  | 562    | 12.28| 585  | 52.44| 484  |
>     | Qwen-Instruct-7B| 54.28 | 313   | 53.43   | 196| 72.70    | 129 | 56.67| 292 | 38.71  | 448    | 12.44| 291  | 48.04| 278  |
>     | Qwen-Instruct-7B[grpo] | 67.55| 504   | 59.52   | 378| 72.30    | 396 | 66.12| 535 | 48.46  | 575    | 14.48| 592  | 54.74| 497  |
>     | AdaThink-Med-Qwen  | **68.34(+0.79)** | 79    | **58.74(-0.78)**   | 109| **73.50(+1.20)**    | 62  | **66.45(+0.33)**| 121  | **48.72(+0.26)**  | 175    | **14.16(-0.32)**| 88  | **54.99(+0.25)**| 106  |
>     | Kimi| 69.99 | 285   | 58.95   | 225| 75.00    | 236 | 66.71| 338  | 46.92  | 376    | 14.89| 330  | 55.41| 298  |
>     | CosFn | 37.94 | 1007  | 49.29   | 919| 75.10    | 885 | 23.32| 998  | 32.82  | 1012    | 10.12| 1020 | 38.10| 974  |
>     | DAST| 65.90 | 260| 59.67 | 327| 69.70    | 153 | 65.86| 205  | 50.51  | 466    | 15.59| 288  | 54.54| 283  |
>     | ShortBetter| 67.94| 333| 58.90| 250| 71.70    | 260 | 65.86| 376  | 46.92  | 405    | 14.36| 387  | 54.28| 335  |

---

> > ### Author Response · Authors · 2025-11-22
> > **Part2**
> >
> > - **Weakness 2 - Senesitivity of Entropy Estimatation**: Thank you for your insightful suggestion. In response to your comment, we have explored the sensitivity of entropy estimation across different LLMs and its impact on compression performance. Specifically, we incorporated a mixture of LLMs, where AdaThink-Med-Llama(+Qwen) refers to utilizing a GRPO-enhanced Qwen2.5-Instruct-7B model to guide the adaptive reasoning of the Llama3.1-Instruct-8B model, leveraging mixed LLMs for comprehensive entropy estimation. Our experimental results demonstrate that combining multiple LLMs leads to more accurate entropy estimates, enabling the adaptive reasoning model to further improve the base model’s performance while maintaining lower inference lengths. This strategy also alleviates the performance drop observed on the MedQA dataset, as mentioned in Weakness 1. Overall, our entropy estimation approach has already shown superior performance with single models, and using mixed LLMs provides more stable entropy estimates, resulting in more efficient pruning of redundant reasoning and refined performance improvement.
> >
> >     | Model    | MedQA Acc. | MedQA Len. | MedMCQA Acc. | MedMCQA Len. | PubMedQA Acc. | PubMedQA Len. | MMLU-ProM Acc. | MMLU-ProM Len. | GPQA-M Acc. | GPQA-M Len. | MedXpert Acc. | MedXpert Len. | Avg Acc. | Avg Len.   |AES|
> >     |------------------------------|------------|------------|--------------|--------------|---------------|---------------|----------------|----------------|-------------|-------------|----------------|----------------|----------|----------|---|
> >     | Llama-3.1-8B-Instruct[cot]   | 68.18 | 461   | 57.32   | 309| 78.20    | 296 | 60.06| 457  | 45.64  | 485    | 16.08| 545  | 54.25    | 425 |-|
> >     | Llama-3.1-8B-Instruct   | 54.75 | 302   | 54.69   | 169| 77.00    | 113 | 57.00| 256  | 35.64  | 345    | 14.48| 342  | 48.93    | 254 |-0.09|
> >     | Llama-3.1-8B-Instruct[grpo]  | 72.50 | 438   | 61.60   | 299| 78.50    | 290 | 63.51| 442  | 46.92  | 461    | 17.26| 529  | 56.72    | 410 |0.17|
> >     | AdaThink-Med-Llama    | 67.00 | 36    | 60.41   | 73 | 78.40    | 54  | 63.58| 85   | 47.94  | 100    | 16.20| 35   | 55.59    | 64    |0.92|
> >     | **AdaThink-Med-Llama(+Qwen)**    | 74.21 | 33    | 62.35   | 60 | 78.22    | 32  | 63.59| 78   | 48.21  | 88| 18.77| 30| 57.56    | 54 |**1.06**|
> >     | Kimi | 70.62 | 145   | 60.93   | 100| 77.50    | 92  | 63.71| 164  | 45.38  | 118    | 15.75| 131  | 55.65    | 125|0.78|
> >     | CosFn| 37.94 | 1007  | 49.29   | 919| 75.10    | 885 | 23.32| 998  | 32.82  | 1012   | 10.12| 1020 | 38.10    | 974|-2.78|
> >     | DAST | 60.64 | 30    | 56.90   | 34 | 76.20    | 39  | 59.48| 71   | 47.69  | 53| 15.35| 82   | 52.71    | 51 |0.74|
> >     | ShortBetter    | 71.48 | 175   | 60.62   | 134| 78.60    | 118 | 62.99| 188  | 45.89  | 219    | 16.08| 171  | 55.94    | 167|0.70|
> >     |&nbsp;|
> >     | Qwen-Instruct-7B[cot]    | 63.86 | 490   | 56.68   | 359| 73.10    | 386 | 62.28| 523  | 46.41  | 562    | 12.28| 585  | 52.44    | 484|-|
> >     | Qwen-Instruct-7B    | 54.28 | 313   | 53.43   | 196| 72.70    | 129 | 56.67| 292  | 38.71  | 448    | 12.44| 291  | 48.04    | 278|0.01|
> >     | Qwen-Instruct-7B[grpo]   | 67.55 | 504   | 59.52   | 378| 72.30    | 396 | 66.12| 535  | 48.46  | 575    | 14.48| 592  | 54.74    | 497|0.11|
> >     | AdaThink-Med-Qwen | 68.34 | 79    | 58.74   | 109| 73.50    | 62  | 66.45| 121  | 48.72  | 175    | 14.16| 88   | 54.99    | 106  |0.93|
> >     | **AdaThink-Med-Qwen(+Llama)**| 69.12 | 61    | 59.87   | 95 | 73.50    | 35  | 66.88| 89   | 48.58  | 143    | 14.33| 75  | 55.38    | 83 |**1.00**|
> >     | Kimi | 69.99 | 285   | 58.95   | 225| 75.00    | 236 | 66.71| 338  | 46.92  | 376    | 14.89| 330  | 55.41    | 298|0.55|
> >     | CosFn| 37.94 | 1007  | 49.29   | 919| 75.10    | 885 | 23.32| 998  | 32.82  | 1012   | 10.12| 1020 | 38.10    | 974|-2.38|
> >     | DAST | 65.90 | 260   | 59.67   | 327| 69.70    | 153 | 65.86| 205  | 50.51  | 466    | 15.59| 288  | 54.54    | 283|0.54|
> >     | ShortBetter    | 67.94 | 333   | 58.90   | 250| 71.70    | 260 | 65.86| 376  | 46.92  | 405    | 14.36| 387  | 54.28    | 335|0.41|

---

> > > ### Author Response · Authors · 2025-11-22
> > > **Part3**
> > >
> > > - **Weakness 3 - Senesitivity of AES**:
> > > Thank you for raising the concern about the sensitivity of the AES metric to hyperparameter choices. We fully understand and appreciate your point. The AES metric is widely adopted in the adaptive reasoning community, as demonstrated in prior works like[1][2][3], and we have used the default parameters from these established studies without any modifications specific to our method. The AES was introduced as a comprehensive measure to balance efficiency gains and performance drops, ensuring fair comparison with existing approaches.
> > >
> > >     In addition to using the standard AES configuration, we have conducted experiments with different hyperparameter settings for AES. The results consistently demonstrate the superiority of AdaThink-Med,validating the robustness and effectiveness of our approach, highlighting its overall advantage in adaptive reasoning tasks.
> > >
> > >     | Model   | avg_acc | avg_len | del_len | del_acc | α = 1, β = 3, γ = 5 | α = 1, β = 5, γ = 5 | α = 1, β = 2, γ = 3 | α = 1, β = 2, γ = 10 | α = 0.5, β = 2, γ = 1 | α = 0.1, β = 1, γ = 2 |
> > >     |-----------------------------|---------|---------|-----------|-----------|---------------------|---------------------|---------------------|-----------------------|-----------------------|-----------------------|
> > >     | Llama-3.1-8B-Instruct[cot]  | 54.25   | 425.27  | -    | -    | -    | -    | -    | - | - | - |
> > >     | Llama-3.1-8B-Instruct  | 48.93   | 254.38  | 0.40 | -0.10 | -0.09    | -0.09    | 0.11    | -0.58 | 0.10| -0.16 |
> > >     | Llama-3.1-8B-Instruct[grpo] | 56.72   | 409.88  | 0.04 | 0.05 | 0.17    | 0.26   | 0.13  | 0.17| 0.11 | 0.05  |
> > >     | Kimi| 55.65   | 124.89  | 0.71 | 0.03 | 0.78    | 0.84    | 0.76    | 0.78| 0.40   | 0.10|
> > >     | CosFn    | 38.10   | 973.64  | -1.29    | -0.30    | -2.78    | -2.78    | -2.18    | -4.27 | -0.94| -0.72 |
> > >     | DAST  | 52.71   | 51.37   | 0.88 | -0.03| 0.74    | 0.74    | 0.79    | 0.60 | 0.41   | 0.03|
> > >     | ShortBetter   | 55.94   | 167.38  | 0.61 | 0.03 | 0.70    | 0.76    | 0.67    | 0.70 | 0.37 | 0.09 |
> > >     | AdaThink-Med-Llama | 55.59   | 63.81   | 0.85 | 0.02 | **0.92**    | **0.97**    | **0.90**    | **0.92** | **0.47** | **0.11**|
> > >     |&nbsp;|
> > >     | Qwen-Instruct-7B[cot]   | 52.44   | 484.08  | -    | -    | -    | -   | -    | -| -    | -    |
> > >     | Qwen-Instruct-7B   | 48.04   | 278.24  | 0.43 | -0.08| 0.01    | 0.01    | 0.17    | -0.41 | 0.13 | -0.13 |
> > >     | Qwen-Instruct-7B[grpo]  | 54.74   | 496.62  | -0.03 | 0.04 | 0.11    | 0.19    | 0.06    | 0.11 | 0.07 | 0.04   |
> > >     | Kimi| 55.41   | 298.09  | 0.38 | 0.06 | 0.55    | 0.67    | 0.50    | 0.55    | 0.31 | 0.10|
> > >     | CosFn    | 38.10   | 973.64  | -1.01| -0.27    | -2.38    | -2.38    | -1.83    | -3.75 | -0.78 | -0.65 |
> > >     | DAST| 54.54   | 283.01  | 0.42 | 0.04 | 0.54    | 0.62    | 0.50    | 0.54 | 0.29    | 0.08 |
> > >     | ShortBetter   | 54.28   | 335.19  | 0.31 | 0.04 | 0.41    | 0.48    | 0.38    | 0.41 | 0.22 | 0.07 |
> > >     | AdaThink-Med-Qwen  | 54.99   | 105.71  | 0.78 | 0.05 | **0.93**    | **1.02**    | **0.88**    | **0.93** | **0.49** | **0.13** |
> > >
> > >     [1]Deepscaler: Surpassing o1-preview with a 1.5b model by scaling rl, 2025b
> > >
> > >     [2]ShorterBetter: Guiding Reasoning Models to Find Optimal Inference Length for Efficient Reasoning
> > >
> > >     [3]O1-Pruner: Length-Harmonizing Fine-Tuning for O1-Like Reasoning Pruning

---

> ### Author Response · Authors · 2025-11-22
> **Part4**
>
> - **Weakness 4 -Compression Strategy Concern**:
>     Thank you for your insightful comment regarding the subjective nature of "excessive length" and the comparison to benchmarks like MATH500 and AIME24. We agree that compression strategies are typically discussed in contexts involving extremely long outputs, such as in mathematical reasoning, where multi-step derivations often require thousands of tokens. However, **in mathematical reasoning, there is often a substantial use of symbols, numbers, and formulas that can significantly increase token consumption**. For example, the tokenization of simple arithmetic equations like "1.0 + 1.0 = 2.0" can result in 11 tokens being used to represent each number and operator, whereas the same statement in **plain text would use fewer tokens**. In contrast, **medical QA tasks, which focus on plain text evidence-based reasoning and differential diagnosis, generally require fewer tokens than math tasks**. It is important to note that public medical QA datasets like **MedQA and PubMedQA contain a significant proportion of simpler questions**, which results in shorter model responses. However, if we were to focus solely on the more difficult questions, we would see longer response lengths comparable to those in benchmarks like MATH500. Our approach is designed to handle such cases effectively, and we have validated its relevance through experiments that demonstrate significant length reduction without sacrificing performance.
>
>     Additionally, to further showcase the adaptability of our method, **we have included experiments with models such as HuatuoGPT-O1 and m1, which generate significantly longer outputs compared to the Llama and Qwen models used in our original paper**. The results, shown in the table below, clearly demonstrate that **our algorithm can effectively optimize reasoning models with longer output lengths**, achieving substantial compression while maintaining or even improving performance. This highlights the **scalability of our method**, making it highly suitable for more complex reasoning tasks that demand longer responses.
>     | Model | MedQA Acc. | MedQA Len. | MedMCQA Acc. | MedMCQA Len. | PubMedQA Acc. | PubMedQA Len. | MMLU-ProM Acc. | MMLU-ProM Len. | GPQA-M Acc. | GPQA-M Len. | MedXpert Acc. | MedXpert Len. | Avg Acc. | Avg Len. | AES |
>     |---------------------------|------------|------------|--------------|--------------|---------------|---------------|----------------|----------------|-------------|-------------|----------------|----------------|----------|----------|------|
>     | HuatuoGPT-o1-8B[cot] | 76.35 | 568   | 62.82   | 445| 79.80    | 445 | 63.71| 521  | 54.35  | 585    | 17.06| 601  | 59.02    | 527 | -    |
>     | Kimi   | 71.01 | 373   | 62.87   | 302| 77.40    | 288 | 67.04| 340  | 50.26  | 401    | 14.49| 363  | 57.18    | 345 | 0.19 |
>     | CosFn  | 68.42 | 260   | 62.63   | 126| 77.30    | 159 | 65.60| 220  | 48.97  | 187    | 14.24| 246  | 56.20    | 200 | 0.38 |
>     | DAST   | 71.17 | 730   | 61.01   | 619| 76.70    | 578 | 64.36| 697  | 45.90  | 794    | 15.02| 765  | 55.69    | 697 | -0.60|
>     | ShortBetter | 71.64 | 336   | 63.18   | 271| 78.40    | 258 | 66.12| 299  | 49.74  | 353    | 14.45| 321  | 57.26    | 306 | 0.27 |
>     | AdaThink-Med    | 72.74 | 183   | 63.54   | 192| 76.60    | 72  | 66.91| 204  | 50.77  | 344    | 13.88| 221  | 57.41    | **203** | **0.48** |
>     |&nbsp;|
>     | m1-7B[cot]  | 75.01 | 2161  | 62.32   | 1749    | 74.20    | 1080| 68.07| 2564 | 51.53  | 3887   | 18.28| 3265 | 58.24    | 2451| -    |
>     | Kimi   | 71.23 | 1425  | 61.26   | 992| 72.50    | 856 | 65.56| 1335 | 50.58  | 2069   | 15.53| 1974 | 56.11    | 1442| 0.23 |
>     | CosFn  | 72.58 | 1566  | 60.57   | 1011    | 72.70    | 774 | 67.99| 1218 | 51.35  | 1655   | 16.23| 1563 | 56.90    | 1298| 0.36 |
>     | DAST   | 70.87 | 2445  | 58.43   | 1995    | 70.10    | 1243| 65.33| 2655 | 48.33  | 4096   | 14.43| 4096 | 54.58    | 2755| -0.44|
>     | ShortBetter | 73.55 | 1359  | 60.58   | 927| 71.40    | 839 | 65.37| 1290 | 50.12  | 1847   | 14.99| 1885 | 56.00    | 1358| 0.25 |
>     | AdaThink-Med   | 74.15 | 982   | 63.35   | 876| 73.10    | 585| 67.99| 1127 | 52.03  | 1596   | 17.55| 1433| 58.03    | **1100**| **0.53** |

---

> ### Author Response · Authors · 2025-11-22
> **Part5**
>
> - **Question 1 and Question 2 - Generalization to Larger and Newer Models**: Thank you for your insightful question. In response, **we have added additional experiments with Qwen-3-32B**, the latest large-scale LLM. Since Qwen-3-32B is a reasoning model fine-tuned with reinforcement learning, we directly applied our adaptive thinking method to this model. The results below demonstrate that our approach consistently achieves greater response length reduction while maintaining a reasonable level of performance degradation, further validating the effectiveness of our method in adapting to newer and larger models.
>
>     | Model    | MedQA Acc. | MedQA Len. | MedMCQA Acc. | MedMCQA Len. | PubMedQA Acc. | PubMedQA Len. | MMLU-ProM Acc. | MMLU-ProM Len. | GPQA-M Acc. | GPQA-M Len. | MedXpert Acc. | MedXpert Len. | Avg Acc. | Avg Len. | AES |
>     |---------------|------------|------------|--------------|--------------|---------------|---------------|----------------|----------------|-------------|-------------|----------------|----------------|----------|----------|-----|
>     | Qwen3-32B| 87.50 | 1679  | 72.34   | 1473    | 73.50    | 722 | 79.67| 1770 | 68.46  | 3009   | 26.41| 2676 | 67.98    | 1888| 0   |
>     | Kimi| 84.23 | 1099  | 69.93   | 1002    | 73.00    | 656 | 79.45| 1335 | 67.25  | 1776   | 25.37| 2108| 66.54    | 1329| +0.19 |
>     | CosFn    | 85.60 | 1523  | 72.12   | 1355    | 72.20    | 585 | 78.53| 1557 | 66.25  | 2553   | 24.45| 2330 | 66.53    | 1651| +0.02 |
>     | DAST| 86.53 | 1345  | 71.88   | 1154    | 73.10    | 724 | 76.53| 1553 | 64.32  | 1889   | 24.32| 2235 | 66.11    | 1513| +0.06 |
>     | ShortBetter   | 84.30 | 1427  | 70.15   | 1238    | 72.50    | 669 | 76.78| 1476 | 67.24  | 1920   | 24.99| 2140 | 65.99    | 1478| +0.07 |
>     | AdaThink-Med  | 86.20 | 1120  | 72.55   | 893| 75.40    | 256 | 77.76| 785  | 66.24  | 1431   | 25.88| 1237 | 67.34    | 954 | **+0.45** |
> - **Question 3 - Results of Closed Source Baselines**: We have added experimental results for **GPT-5 and Gemini-2.5-Pro**, which are trained on significantly larger datasets and feature much larger model sizes (hundreds of billions of parameters). For the AES calculations, we used the results from the Qwen2.5-Instruct-7B model after fine-tuning with GRPO as a reference. It is important to note that **GPT-5 has been optimized with OpenAI’s adaptive reasoning algorithm**. The results below show that while GPT-5 and Gemini-2.5-Pro achieve impressive performance, Gemini-2.5-Pro generates more redundant reasoning outputs, leading to suboptimal inference efficiency. In comparison, **our model based on the Llama backbone achieves inference efficiency comparable to that of GPT-5**. While performance is slightly lower due to model size limitations (8B parameters), the overall inference efficiency sufficiently demonstrates the effectiveness of our approach in balancing performance and computational cost.
>
>     | Model    | MedQA Acc. | MedQA Len. | MedMCQA Acc. | MedMCQA Len. | PubMedQA Acc. | PubMedQA Len. | MMLU-ProM Acc. | MMLU-ProM Len. | GPQA-M Acc. | GPQA-M Len. | MedXpert Acc. | MedXpert Len. | Avg Acc. | Avg Len. | AES  |
>     |--------------------|------------|------------|--------------|--------------|---------------|---------------|----------------|----------------|-------------|-------------|----------------|----------------|----------|----------|------|
>     | Qwen-GRPO| 67.55 | 504   | 59.52   | 378| 72.30    | 396 | 66.12| 535  | 48.46  | 575    | 14.48    | 592  | 54.74    | 497 | - |
>     | GPT-5    | 76.07 | 239   | 62.89   | 198| 68.00    | 278 | 72.42| 316  | 59.13  | 320    | 35.25| 335  | 62.46    | 281 | **+0.86** |
>     | Gemini-2.5-Pro| 74.20 | 505   | 62.60   | 408| 70.80    | 220 | 68.12| 549  | 57.32  | 549    | 27.91| 612  | 60.16    | 474 | +0.34 |
>     | AdaThink-Med-Llama   | 67.00 | 36    | 60.41   | 73 | 78.40    | 54  | 63.58| 85   | 47.94  | 100    | 16.20    | 35   | 55.59    | 64  | +0.92 |
>     | AdaThink-Med-Qwen   | 68.34 | 79    | 58.74   | 109| 73.50    | 62  | 66.45| 121  | 48.72  | 175    | 14.16| 88   | 54.99    | 106 | +0.80 |

---

### Author Response · Authors · 2025-11-27
**Kind reminder of the discussion**

We sincerely thank all reviewers and ACs for your constructive and detailed reviews. We have answered the reviewers' questions one by one, and revised our paper according to the suggestions, denoted in red in our revised paper. We sincerely appreciate that Reviewer KxCJ has read our response and raised the score, and Reviewer 52Pp has maintained their positive score. We would like to know if our response has addressed your concerns and questions. If you have any further concerns or suggestions for the paper or our rebuttal, please let us know. We would be happy to engage in further discussion and manuscript improvement.

---

### Author Response · Authors · 2025-12-03
**Summary Response [Rebuttal Overview & Resolution of Primary Concerns] Part 1/3**

Dear AC, SAC, and PC Members,

We are sorry to learn about the recent information leak on OpenReview and the resulting impact on the reviewing process of ICLR. We understand that this unexpected incident has introduced additional workload, and we would like to express our sincere appreciation to the AC for the extra effort required during this challenging situation. To help reduce the burden on the AC, we summarize below the key progress that was explicitly acknowledged by the reviewers during the two-week rebuttal period prior to the information leak incident.

Specifically, we have incorporated the additional ablation studies and quantitative metrics suggested by the reviewers. **These experiments provide a more explicit verification of AdaThink-Med, reinforcing the efficacy already demonstrated in our original submission**. We respectfully believe that the current ratings may not fully reflect the strength and completeness of our work.

***1. Reviewer Comments: Strong Consensus on Motivation & Results***

Despite the mixed ratings, there is a strong consensus on the contribution of our framework and the strength of our results. We quote the key positive feedback from each reviewer as follows:

- 52Pp: "good adaptive thinking framework", "dynamic mode switching without separate classifiers", "rigorous and comprehensive experiments", "addresses a critical bottleneck for medical LLM deployment"
- KxCJ: "principled difficulty signal", "prevents collapse", "substantial efficiency gains", "beats strong length‑calibration baselines", "emergent non‑thinking / thinking modes without extra supervision", "good generalization"
- xhiU: "strong motivation", "achieves the best AES", "scaling from hundreds of tokens to a few dozen", "thinking mode could help influence similar training strategies"
- B55m: "clearly formulated and technically well-motivated", "systematically constructed", "comprehensive empirical analysis"


As is evident, the reviewers unanimously affirm **the value of the AdaThink-Med framework design** and the **superior empirical performance**. The main concerns of the reviewers stem from some detailed questions about the paper and the supplementation of ablation experiments.

***2. Rebuttal and Discussion Phase***

We are deeply encouraged that the two reviewers who responded to our rebuttal (Reviewer KxCJ and Reviewer 52Pp) have acknowledged that our detailed responses and additional experiments have addressed their main concerns. **Reviewer KxCJ raised the score from 4 to 6, and Reviewer 52Pp confirmed that the concerns were resolved and maintained a positive score of 6**. Unfortunately, the other two reviewers, xhiU and B55m, **did not provide any response**. Below is a brief summary of the review history focusing on the main concerns raised by each reviewer.

## Reviewer 52Pp (Initial Rating:6 -> Final Stance: Maintain Positive)
- *Initial Concerns*:
    1. The robustness to low-quality or biased candidate outputs.
    2. It does not explore how medical domain characteristics (e.g., clinical ambiguity, multi-step differential diagnosis, rare diseases) affect difficulty estimation.
    3. It lacks direct comparison with state-of-the-art medical LLMs optimized for efficiency.
    4. AdaThink-Med is limited to text-based QA.
    5. The paper does not explain how AdaThink-Med handles edge cases in medical QA.


- *Our Responses*:

    1. We conducted additional experiments on low-accuracy, low-quality datasets to explore the model’s robustness under such extreme scenarios. The results demonstrate that the model consistently achieves improved efficiency optimization.
    2. We clarify that our method is general in nature—while we focused our investigation on the medical QA domain, **the proposed adaptive training approach itself is novel for both general and medical fields**. We further **explain the pressing efficiency requirements in real-world medical deployment scenarios**, and we acknowledge the reviewer’s suggestions as valuable directions for future exploration.
    3. We emphasize that **our work presents the first end-to-end medical adaptive reasoning LLM**. Since the non-end-to-end baseline SynapseRoute is not publicly available, a direct comparison could not be performed.
    4. We note that **our research centers on Medical LLMs rather than Vision-Language Models (VLMs)**. Nevertheless, **we have supplemented our study with experiments on a VLM-based radiology report generation task**, in which our model achieves optimal performance in terms of content completeness and response conciseness.
    5. We further test AdaThink-Med’s robustness in two edge cases: whether it produces verbose outputs for hard questions or misses key reasoning steps when reducing tokens.

- *Reviewer's Final Feedback*:
    > I believe the authors have **solved most of my problems**; I shall refer to the comments of other reviewers to **maintain my positive score**.

---

> ### Author Response · Authors · 2025-12-03
> **Summary Response [Rebuttal Overview & Resolution of Primary Concerns] Part 2/3**
>
> ## Reviewer KxCJ (Initial Rating:4 -> Final Stance: 6)
> - *Initial Concerns*:
>     1. AES sensitivity & cost realism.
>     2. Batch‑wise normalization can introduce instability or dependence on batch composition.
>     3. Inference‑time mechanism.
>     4. Dataset‑selection experiment lacks controls, e.g. random subsets experiments.
>     5. Open‑ended RL setup.
>     6. Clarify whether lengths are normalized or reported per‑model tokenizer.
>
> - *Our Responses*:
>
>     1. We first clarify that **the AES metric is the community-adopted default**, and we have employed consistent parameter configurations throughout. In response to the reviewer's request, **we have supplemented the analysis with parameter sensitivity tests, the results of which consistently demonstrate the optimal efficiency of AdaThink-Med**. Regarding cost realism, we explain that under the same model architecture, comparing based on output tokens provides a valid measure of cost. Accordingly, as suggested, we have presented the main results separately for different tokenizers.
>     2. We explain that **the batch min-max strategy was introduced to enhance training stability**. We have supplemented experiments with and without this strategy, as well as tests with varying batch sizes. The results show that omitting batch min-max compromises training stability, **while different batch sizes affect convergence speed but not the final performance**, highlighting the robust performance of AdaThink-Med.
>     3. We clarify that after adaptive training, **the model spontaneously acquires the ability to generate responses of appropriate length based on the input question**.
>     4. We have added experiments using randomly selected subsets of the same scale, the results of which demonstrate the high efficiency of AdaThink-Med’s data selection capability.
>     5. We provide a detailed supplementary explanation of the parameter settings for open-ended adaptive training and the training methodology for the verifier.
>     6. Adopting the reviewer's insightful suggestion, we have reported the results separately for each tokenizer.
>
> - *Reviewer's Final Feedback*:
>     > ··· **The quantitative experiments have addressed my earlier concerns**··· **I am inclined to increase my score to a 6**···
>
>     Subsequently, we held an in-depth discussion with the reviewer concerning the paper's contributions. The reviewer responded positively, offering extensive constructive feedback and even assisting in summarizing the four core contributions of our work. The reviewer concluded with the following remarks:
>
>     > ···**That said, I appreciate the thoroughness of the experimental section and have increased my score accordingly.**···
>
> ## Reviewer xhiU (No Response)
> - *Initial Concerns*:
>
>     1. Decreased performance by over 6 points across several benchmarks.
>     2. It is unclear how deriving responses from mixtures or LLMs or a wider breadth of LLMs would affect this strategy.
>     3. The AES seems very sensitive to choice of hyper-parameters.
>     4. While "excessive length" is subjective, most competitors in Table 1 are generating only a few hundred tokens.
>
> - *Our Responses*:
>
>     1. **We respectfully clarify that the reviewer appears to have misinterpreted the experimental results**. In reality, **only the LLama-Instruct-8B model exhibits a performance drop of 5.5 percentage points on MedQA, while the declines on other benchmarks are generally within 1 percentage point**. Notably, on the GPQA-M dataset, there is even a performance gain of 1.02 percentage points. This contrasts with the reviewer's claim of "over 6 points across several benchmarks."
>
>     2. We first reiterate that our research primarily focuses on adaptive inference optimization for dense LLMs. As requested by the reviewer, we have supplemented experiments with a mixture of LLMs. The results demonstrate more accurate uncertainty estimation, which further enhances the efficiency gains of AdaThink-Med and also resolves the performance drop observed on the MedQA dataset.
>
>     3. Consistent with the concern raised by Reviewer KxCJ, we explain that the evaluation metrics follow the community’s default configurations. We have also added hyperparameter sensitivity experiments, which confirm the consistently superior performance of AdaThink-Med.
>
>     4. We first explain that **mathematical reasoning tasks typically involve longer token sequences due to the tokenization of numerous symbols, compared to plain text reasoning tasks**. Secondly, we note that public medical QA datasets contain a significant proportion of **simple questions, leading to shorter average response lengths**, although responses spanning thousands of tokens do occur. Furthermore, **we have supplemented adaptive optimization experiments on much longer reasoning models, specifically HuatuoGPT-o1 (527 tokens) and m1 (2451 tokens)**. The results highlight the excellent compression efficiency achieved by AdaThink-Med.

---

> > ### Author Response · Authors · 2025-12-03
> > **Summary Response [Rebuttal Overview & Resolution of Primary Concerns] Part 3/3**
> >
> > ## Reviewer B55m (No Response)
> > - *Initial Concerns*:
> >
> >     1. It is unclear if the approach addresses any challenges unique to medical reasoning, or if it simply applies a general technique to medical datasets.
> >     2. There is insufficient analysis of potential failure cases or model sensitivity to hyperparameter choices and data distribution shifts.
> >     3. The proposed framework relies on model-based uncertainty scores to define question difficulty, which may not align with human or domain-expert judgments.
> >     4. This paper lacks medical professionals (e.g., clinicians, medical students) for qualitative or quantitative assessment.
> >     5. Potential side effects analysis. Such as generating unnecessarily verbose outputs for hard questions, or reducing token usage while missing important reasoning steps.
> >
> > - *Our Responses*:
> >
> >     1. We highlight that **our entropy-based adaptive inference difficulty metric is novel in both general and medical domains**, and not merely an application of existing techniques to medical data. **Our focus on medical applications is driven by both our research direction and the pressing need for clinically deployable solutions that address real-world deployment challenges**.
> >
> >     2. We have added an analysis of failure cases in Section 4.1 (Figure 5), along with experiments on model sensitivity to hyperparameters (TOP-K, temperature T) and efficiency optimization under open-ended data distributions, further demonstrating the robustness of AdaThink-Med.
> >
> >     3. We clarify that adaptive reasoning can be improved via two main approaches: a model-based method (relying on the model's own difficulty estimation) and a human-based one (incorporating human preferences). **Our work focuses on the model-based approach due to its superior scalability, as the human-based method requires extensive preference annotation**.
> >
> >     4. We conducted a user-centric evaluation involving three clinicians who assessed 500 randomly selected samples. AdaThink-Med received significantly higher ratings than other methods in terms of response accuracy and acceptable reasoning redundancy, indicating its superior generation quality as perceived by human evaluators.
> >
> >     5. Additional experiments detail the robustness of AdaThink-Med in two edge cases: avoiding overly verbose outputs for hard questions, and preserving essential reasoning steps while reducing token usage.
> >
> > ## Conclusion
> > The two reviewers (KxCJ and 52Pp) who submitted their final comments have reached a clear consensus that our rebuttal has successfully addressed their initial concerns. The core strengths of our work—significant compression of reasoning cost with only minor or even improved performance, supported by extensive quantitative, qualitative, and human evaluations were validated throughout this process.
> >
> > Regarding the two reviewers (B55m and xhiU) who did not respond, their concerns primarily focused on requests for clarifications on certain details, experiments on model generalizability, and robustness tests of hyperparameters. We believe our thorough and detailed supplementary experiments adequately resolve these issues. We sincerely thank all reviewers for their insightful comments, which have undoubtedly strengthened the paper.
> >
> > We hope the Area Chair will take into account **the positive response** of the reviewers who engaged in the discussion and their full acknowledgment of our supplementary evidence. **We also trust that our comprehensive experiments will provide a convincing response to the points raised by the non-responsive reviewers**. Thank you for your time and consideration.
> >
> > Once again, we would like to extend our best gratitude for your dedicated efforts.
> >
> > Best regards,
> >
> > The Authors

---

### Meta-Review · Area_Chair_rDFh · 2026-01-06

**Summary:**

This paper presents AdaThink-Med, an end-to-end reinforcement learning framework designed to enhance the adaptive reasoning capabilities of medical large language models by calibrating response length based on estimated question difficulty. Reviewers acknowledged the work's strong motivation and the substantial reduction in output tokens achieved across several benchmarks. However, key concerns were raised regarding a non-trivial drop in accuracy on certain datasets, limited evidence that the method addresses challenges unique to medical reasoning beyond applying a general technique to medical data, and questions about the novelty of its core mechanism in light of existing entropy-guided reasoning approaches. While the authors provided extensive rebuttals and additional experiments, the fundamental issues of performance trade-offs and domain-specific contribution remained insufficiently resolved. Given these persistent limitations, the paper is not recommended for acceptance at this stage.

**Reviewer Concerns:**

The authors' rebuttal provided substantial additional experiments and clarifications that successfully addressed several specific technical concerns raised by the reviewers. Specifically, concerns regarding the sensitivity of the AES metric to hyperparameters, the robustness of the batch normalization and entropy estimation, the need for control experiments in dataset selection, the model's performance on low-quality data and edge cases, and its generalization to larger models and different tasks (e.g., radiology report generation) were all met with new data and analysis. Reviewers 52Pp and KxCJ explicitly acknowledged that their primary concerns were resolved, leading one to maintain a positive score and the other to increase their score.

Despite the comprehensive rebuttal, fundamental concerns regarding the paper's core contribution and domain-specific novelty remain unresolved. A key criticism, particularly highlighted by Reviewer KxCJ, is that the central technique of using entropy to guide reasoning length is not novel, as evidenced by several contemporaneous works. The rebuttal did not convincingly counter the argument that the method is primarily a straightforward application of an emerging general technique to medical datasets, without introducing a medically-specific innovation. Related to this, concerns about whether the "difficulty" estimation aligns with clinical or expert judgment, rather than just model uncertainty, were not adequately addressed. The clinical evaluation, while added, remains limited in scale. Furthermore, the significant accuracy drop on the MedQA benchmark for one model variant, even if an outlier, points to potential instability in the trade-off between efficiency and accuracy that was not fully mitigated. These unresolved issues pertain to the paper's positioning and claimed novelty within the field.

**Reviewer Scores:**

| Reviewer | Initial Score | Predicted New Score | Reason |
|-|-|-|-|
| KxCJ | 4 | 6 | Explicitly raised score to 6 after discussion. Extensive additional experiments (sensitivity, batch norm, inference) addressed technical concerns. |
| 52Pp | 6 | 6 | Maintained positive stance, confirming most concerns were resolved by the rebuttal and additional experiments. |
| xhiU | 4 | 4 | The core concern about justifying the accuracy-length trade-off (notable drop on MedQA) was not fully alleviated. |
| B55m | 4 | 4 | The fundamental concern about the lack of a clear, domain-specific medical contribution (vs. applying a general technique) remained unresolved, even with the added clinician evaluation. |

---

### Decision · Program_Chairs · 2026-01-26

Reject